# Beyond Real-world Benchmark Datasets: An Empirical Study of Node Classification with GNNs

**Seiji Maekawa[1], Koki Noda[2], Yuya Sasaki[1], Makoto Onizuka[1]**
[1]Graduate school of Information Science and Technology, Osaka University
[2]TDAI Lab Co., Ltd.
{maekawa.seiji,sasaki,onizuka}@ist.osaka-u.ac.jp, koki.noda@tdailab.com

## Abstract

Graph Neural Networks (GNNs) have achieved great success on a node classification task. Despite the broad interest in developing and evaluating GNNs, they have been assessed with limited benchmark datasets. As a result, the existing evaluation of GNNs lacks fine-grained analysis from various characteristics of graphs. Motivated by this, we conduct extensive experiments with a synthetic graph generator that can generate graphs having controlled characteristics for fine-grained analysis. Our empirical studies clarify the strengths and weaknesses of GNNs from four major characteristics of real-world graphs with class labels of nodes, i.e., 1) class size distributions (balanced vs. imbalanced), 2) edge connection proportions between classes (homophilic vs. heterophilic), 3) attribute values (biased vs. random), and 4) graph sizes (small vs. large). In addition, to foster future research on GNNs, we publicly release our codebase that allows users to evaluate various GNNs with various graphs. We hope this work offers interesting insights for future research.

## 1 Introduction

The semi-supervised node classification task is one of the hottest topics. Its goal is to predict unknown labels of the nodes by using the topology structure and node attributes, given partially labeled networks. In the last several years, graph neural networks (GNNs) [3, 9, 12, 16, 17, 22, 23, 29, 34, 38, 45, 46] have gained wide research interest from various domains including chemistry [7, 10], physics [27], social science [16], and neuroscience [43]. For further improvement of classification accuracy, most researches focus on developing expressive and powerful GNNs.

Towards practical use cases of GNNs, researchers and developers need to deeply understand the strengths and weaknesses of GNNs from various aspects such as their accuracy, efficiency, and parameter sensitivity. To this end, it is important to conduct extensive experiments using various graphs with different characteristics.

**Limitations of existing evaluation of GNNs.** Several studies [8, 10] addressed benchmarking the performance of GNNs. However, the comprehensive evaluation of GNNs has been challenging since most models are assessed on well-known but limited benchmark datasets such as Cora, Citeseer, and PubMed [36]. Although recent studies [14, 19] provide a collection of real-world datasets to mitigate the shortage of datasets, these datasets are still insufficient to understand the pros/cons of various model architectures. Concretely, we require a variety of graphs to evaluate GNNs, in terms of four major characteristics of attributed graphs with class labels of nodes: 1) class size distributions, 2) the relationship between classes and topology, 3) the relationship between classes and attributes, and 4) graph sizes.

To examine GNNs on graphs with different characteristics, a study [8] uses a commonly used graph generator, SBM [1]. However, the quality and variety of synthetic graphs generated by SBM are

limited since it cannot generate realistic graphs and does not support generating node attributes. Due to the limitation, existing benchmarking studies of GNNs using real-world and synthetic graphs lack fine-grained analysis, e.g., the evaluation on graphs by changing one or a few target characteristic(s) while keeping other characteristics unchanged. To the best of our knowledge, no studies conducted comprehensive evaluations of GNNs on various synthetic graphs with controlled characteristics regarding classes despite the great interest in developing and evaluating GNNs.

**Contributions.** To understand the pros/cons of GNNs, we empirically study the performance of GNNs through extensive experiments on various graphs by synthetically changing one or a few target characteristic(s) of graphs, while real-world benchmark datasets cannot provide fine-grained analysis due to their limited variety. For fine-grained analysis of GNNs, we adopt GenCAT, the state-of-the-art graph generator [21] that can more flexibly control the characteristics of generated graphs with node labels than traditional generators such as SBM. In terms of the above four major characteristics, 1) class size distributions, 2) the relationship between classes and topology, 3) the relationship between classes and attributes, and 4) graph sizes, we attempt to answer the following questions: **Q1.** *How largely do class size distributions affect the performance of GNNs?* **Q2.** *How effectively do GNNs work on graphs with various edge connection proportions between classes?* **Q3.** *How largely do attribute values contribute to the performance of GNNs?* **Q4.** *How effectively do GNNs work on graphs with various sizes?*

We summarize key takeaways of our empirical study, which we hope could benefit the community focusing on developing new GNN algorithms:

- GNNs including the state-of-the-art methods suffer from a class imbalance problem that typically deteriorates the performance of multi-class classification. Interestingly, the simplest algorithm SGC [32] outperforms recent complicated GNN algorithms in a class imbalanced setting because the complicated algorithms tend to over-fit to major classes.
- GNNs generalizing to graphs with few edges within each class (we call heterophilic graphs) provide marginal classification performance gains in a heterophily setting from a graph-agnostic classifier MLP. This indicates that such GNNs almost ignore the topology information in the setting.

We hope the use of our framework will greatly relief the burden of comparing existing baseline GNNs and developing new algorithms. Our codebase[1] is available under the MIT License.

## 2 Preliminaries

**Notation.** An *undirected attributed graph with node labels* is a triple $G = (\boldsymbol{A}, \boldsymbol{X}, \boldsymbol{C})$ where $\boldsymbol{A} \in \{0,1\}^{n \times n}$ is an adjacency matrix, $\boldsymbol{X} \in \mathbb{R}^{n \times d}$ is an attribute matrix assigning attributes to nodes, and a class matrix $\boldsymbol{C} \in \{0,1\}^{n \times y}$ contains class information of each node, and $n, d, y$ are the numbers of nodes, attributes, and classes, respectively. We call a set of nodes with the same label a *class* and define $\Omega_l$ as the set of nodes labeled with $l$.

**Problem Definition (Node classification).** We split nodes into train/validation/test sets. Given an adjacency matrix $\boldsymbol{A}$, an attribute matrix $\boldsymbol{X}$, and a partial class matrix $\boldsymbol{C}'$ which contains class information of nodes in the train and validation sets, we predict the labels of the nodes in the test set.

## 3 A General Summary of Graph Neural Networks for Node Classification

Graph convolutional operation is inarguably the hottest topic in graph-based deep learning. Inspired by CNNs, GNNs using multiple convolutional layers learn local and global structural patterns. Many methods based on graph convolution networks have been proposed. We briefly explain several key instances of GNNs.

### 3.1 Typical Convolutional GNNs

Multi-layer graph convolutional network (`GCN`) is a standard graph convolutional network model which was proposed in [16]. The graph convolution is closely related to spectral analysis on graphs

---

[1] `https://github.com/seijimaekawa/empirical-study-of-GNNs`

that has a solid mathematical foundation in graph signal processing. Graph signal processing that typically uses eigen-decomposition requires $O(n^3)$ computational complexity. To reduce the complexity, `ChebNet` [4] approximates the eigen-decomposition by using $k$-th power Chebyshev polynomials, where $k$ indicates the number of hops for local feature propagation. Mixture model network (`MoNet`) [23] introduces a different approach to assign weights to adjacent nodes by using the relative position between every node and its neighbor. Several existing approaches such as GCN [16] can be generalized as instances of MoNet. Graph attention network (`GAT`) [29] adopts attention mechanisms to learning the relative weights between each pair of connected nodes. Also, GAT increases its expressive capability by using the multi-head attention mechanism. Jumping knowledge network (`JKnet`) [35] introduces skip connections between different convolution layers. `SGC` [32] simplifies GCN by removing nonlinearities between convolutional layers. SGC exhibits comparable performance to GCN while being computationally more efficient and fitting fewer parameters. Graph isomorphism networks [34] have a simple architecture that is provably expressive and is as powerful as the Weisfelier-Lehman graph isomorphism test.

**Sampling-based GNNs.** Sampling-based GNNs [12, 38, 39] compute node representations from enclosing subgraphs of an input graph, i.e., neighborhood samplings. `GraphSAGE` [12] uniformly samples a fixed number of neighbors for each node. `GraphSAINT` [39] samples subgraphs and propagates node embeddings within the subgraphs. It learns good representations of a whole graph by combining information of many subgraphs together. `Shadow-GNN` [38] decouples the number of GNN layers and the scope of subgraph extraction. This allows Shadow-GNN to propagate node embeddings within many hops while keeping the sizes of subgraphs small. Hence, it achieves the state-of-the-art classification accuracy with lower hardware cost than other existing methods.

### 3.2 GNNs Generalizing to Heterophilic Graphs

Many GNN models implicitly assume homophily in graphs, i.e., nodes having the same class label tend to connect to each other. However, in the real world, there are also other settings where "opposites attract", leading to graphs with heterophily, i.e., connected nodes are likely from different classes or have dissimilar features. For example, fraudsters are more likely to connect to accomplices than to other fraudsters in online purchasing networks.

While most existing GNNs fail to capture the heterophily in graphs, recent works [3, 22, 46] have addressed generalizing homophily and heterophily settings. To capture the heterophily property, `H2GCN` [46] combines three designs; 1) ego- and neighbor-embedding separation, 2) higher-order neighborhoods, and 3) combination of intermediate representations. Thanks to these designs, H2GCN can increase the representation power of GNNs in the challenging settings with heterophily and achieves better results than existing methods ignoring the heterophily property. `FSGNN` [22] flexibly utilizes information from high-order neighbors while reducing the number of model parameters by decoupling the depth of feature propagation and the number of layers of neural networks. Also, FSGNN adopts feature selection parameters weighting important hops for a node classification task so that the model can perform well on graphs with the heterophily property. `GPRGNN` [3] automatically learns generalized PageRank parameters associated with each step of feature propagation. The parameters depend on the contributions of different steps during the information propagation, similarly to the feature selection parameters of FSGNN. `LINKX` [18] is a simple and scalable method based on MLP, which achieves state-of-the-art performance for learning graphs with the heterophily property (called non-homophilious graphs in the paper). However, LINKX obtains lower accuracy on graphs with the homophily property than existing GNNs since it does not adopt graph convolution that is tailored to capture the homophily.

## 4 Real-world Benchmark Datasets and Their Limitations

The benchmark datasets have played an important role for fair comparisons between new GNN algorithms and existing algorithms. In this section, we first summarize benchmark datasets. Then, we discuss their limitations to fine-grained analysis of GNNs.

### 4.1 Benchmark Datasets

**Homophilic graphs.** There are three widely used citation networks (`Cora`, `Pubmed`, and `Citeseer` [36]). Also, Amazon co-purchase networks for specific domains (`Computers` and `Photo`) are used [3, 28]. `Reddit` [12] is a social network dataset where nodes represent posts and two posts are connected if the same user left comments on them. A collection of large homophilic graphs is released recently by the Open Graph Benchmark project [14], including two citation networks (`ogbn-arxiv` and `ogbn-papers100M`) and an Amazon product co-purchasing network (`ogbn-products`).

**Heterophilic graphs.** `Texas`, `Wisconsin` and `Cornell` are graphs representing links between web pages of the corresponding universities[2]. `Actor` [25] is a co-occurrence network in which each node corresponds to an actor and edges indicate their co-occurrence on the same Wikipedia page. `Chameleon` and `Squirrel` [25] are Wikipedia networks on specific topics. A series of large heterophilic graphs from diverse domains is released by a recent study [18, 19]. It consists of five online social networks (`Penn94`, `Pokec`, `genius`, `Deezer-Europe`, and `Twitch-Gamers`), two citation networks (`arXiv-year` and `snap-patents`), a Wikipedia web page network (`Wiki`), and a hotel and restaurant review dataset (`YelpChi`)[3].

### 4.2 Limitations of Current Evaluation of GNNs

Node classification results depend on four major characteristics of graphs having node labels: 1) class size distributions (balanced vs. imbalanced), 2) edge connection proportions between classes (homophily vs. heterophily), 3) attribute values (biased vs. random), and 4) graph sizes (small vs. large). Though several real-world graphs were publicly released, evaluations using them suffers from limitations with respect to performing fine-grained analysis, i.e., it is hard to investigate how largely each single characteristic of given graphs (e.g., the class, topology, and attribute information) affects node classification results because the real-world graphs are typically different from each other in multiple characteristics. Such deep analysis and understandings of GNN algorithms are helpful to enhance developing new GNNs that have gained large attentions.

## 5 Synthetic Graph Generator

Since we aim to address fine-grained analysis on GNNs for node classification, graph generators that we adopt need to satisfy two requirements; 1) generating graphs in which nodes are associated with attributes and labels, and 2) flexibly controlling the characteristics of generated graphs.

We adopt `GenCAT` graph generator [21], the only method satisfying the above two requirements. Current state-of-the-art methods [1, 30] fail to support either of the requirements satisfied by GenCAT. A recent work `FastSGG` [30] efficiently generates trillion-scale graphs but cannot flexibly control edge connections between classes, i.e., it fails to capture the second characteristic discussed in Section 4.2. Another example is `SBM` [1] which does not generate graphs similar to real-world graphs as shown in [21] and does not support attribute generation, i.e., it fails to capture the third characteristic. Also, cSBM [6] is used to generate synthetic graphs to evaluate GNNs in [3]. However, cSBM focuses on the simple case where there are two classes in a graph. Hence, it is not suitable for comprehensive evaluation since real-world graphs have various numbers of classes.

GenCAT is the state-of-the-art attributed graph generator that allows users to flexibly control the four characteristics of generated graphs which we discussed in Section 4.2. Since GenCAT captures the relationship between classes, attributes, and topology, the attributes and topology in generated graphs share the class structure. Also, GenCAT can simulate the existing generator SBM without any modifications of GenCAT itself. Please see more detailed and precise procedures in [21].

### 5.1 Major Inputs of GenCAT Graph Generator

To capture various characteristics in attributed graphs, GenCAT inputs several parameters called class features and graph features. To reduce the effort of users, GenCAT provides a function extracting the parameters from a given graph with node labels.

---

[2]http://www.cs.cmu.edu/afs/cs.cmu.edu/project/theo-11/www/wwkb/
[3]These datasets are called non-homophilious datasets in the papers [18, 19]

**Class features.** First, we introduce *class size distribution* $\rho \in \mathbb{R}^k_+$ such that $\rho_l = |\Omega_l|$ for each label $l$. Next, we introduce *class preference mean* $M \in \mathbb{R}^{k \times k}$ representing the edge connection proportions between classes, to simulate homophily/heterophily phenomena. We formulate class preference mean between class $l_1$ and class $l_2$ as follows:

$$M_{l_1 l_2} = \frac{1}{|\Omega_{l_1}|} \sum_{i \in \Omega_{l_1}} \left( \sum_{j \in \Omega_{l_2}} A_{ij} / \sum_{j=1}^{n} A_{ij} \right). \tag{1}$$

Then, we introduce *attribute-class correlation* $H \in \mathbb{R}^{d \times k}$ indicating the strength of the correlation between attributes and classes. We formulate attribute-class correlation between attribute $\delta$ and class $l$ as the average value of attribute $\delta$ of nodes in class $l$:

$$H_{\delta l} = \frac{1}{|\Omega_l|} \sum_{i \in \Omega_l} (X_{i\delta}). \tag{2}$$

**Graph features.** There are several fundamental statistics of graphs such as the numbers of nodes, edges and attributes. By inputting them, GenCAT can generate graphs with various sizes.

## 6 Empirical Studies

We aim to clarify the strengths and weaknesses of GNNs from viewpoints of four major characteristics of graphs with node labels, i.e., 1) class size distributions, 2) edge connection proportions between classes, 3) attribute values, and 4) graph sizes. First, we describe the setups of our empirical studies. Second, we show experiments regarding node classification performance and discuss the results in detail. Finally, we measure the training time per epoch of GNNs on synthetic graphs with various sizes to investigate their efficiency[4].

**Experimental Setup.** We use 16 GNNs including the state-of-the-art methods such as GPRGNN [3] and shaDow-GNN [38] as we discussed in Section 3[5]. We also execute a graph-agnostic classifier (i.e., it ignores the topology structure), multi-layer perceptron (MLP), in order to evaluate how largely the topology structure contributes to classification performance. We use random class splits (60%/20%/20% of nodes for train/validation/test) whose ratio is provided by [3]. We perform a grid search to tune the hyperparameters of GNNs to generated graphs for each setting. The hyperparameter search space and their best parameter set for each setting are reported in our codebase for the experimental reproducibility. To reduce the randomness, we generate three graphs for each setting of graph generation and execute GNNs with three restarts for each generated graph. We report average scores and standard deviations as error bars. In our experiments, we extract parameters of the graph generation from the Cora dataset, i.e., we obtain class and graph features from Cora. Note that Cora has 2708 nodes, 5278 edges, 1433 attributes, and 7 classes. Then, we configure one or a few parameters for each setting[6]. We use f1-macro to evaluate classification quality, which can better reflect the performance on minority classes than accuracy. We measure training time on a NVIDIA Tesla V100S GPU (32GB) and Intel(R) Xeon(R) Gold 5220R CPU×2 (378GB).

### 6.1 Classification Quality on Synthetic Graphs with Various Characteristics

#### 6.1.1 Various Class Size Distributions

Most studies on GNNs for node classification do not consider class size distributions in their evaluations. In fact, existing studies use only accuracy to evaluate the classification quality since they do not consider class size distributions. However, class size distributions are typically imbalanced [13] in practical use cases. Instead of using accuracy, we use f1-macro which is suitable to equally treat major and minor classes. The trade-off between fairness and accuracy has been actively discussed [2, 5]. We also show experimental results regarding accuracy in the supplementary materials.

---

[4]Since total execution time is the product of training time per epoch and the number of epochs until convergence, it highly depends on the criteria for early stopping and learning rate. Hence, we measure training time per epoch and discuss it since it is difficult to directly analyze the total execution time.

[5]We drop several existing methods such as GIN [34] and SIGN [9] due to the space limitation. However, we do not fail to use the state-of-the-art methods.

[6]Parameters not mentioned are set to those extracted from Cora.

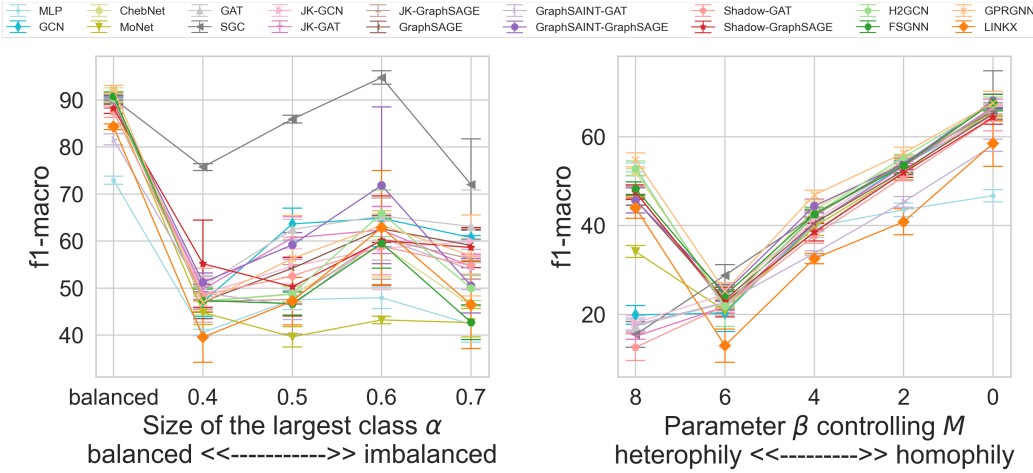

Figure 1: Classification performance on graphs with various class sizes.

Figure 2: Classification performance on graphs with various edge connection proportions.

**Detailed setup.** To investigate how largely class imbalance affects the classification performance, we conduct experiments on synthetic graphs with configured class size distributions $\rho^{\text{conf}}$ as follows:

$$\rho_l^{\text{conf}} = \begin{cases} \alpha & (l = 1) \\ \alpha * \left(1 - \sum_{l'=1}^{l-1} \rho_{l'}^{\text{conf}}\right) & (1 < l < y) \\ 1 - \sum_{l'=1}^{l-1} \rho_{l'}^{\text{conf}} & (l = y) \end{cases}, \tag{3}$$

where $\alpha$ indicates the size of the largest class. We set $\alpha \in [0.4, 0.5, 0.6, 0.7]$ to investigate how GNNs perform on graphs with imbalanced classes, which most existing studies ignore. Note that since the Cora dataset has seven classes, the classes in generated graphs are imbalanced when $\alpha = 0.4, 0.5, 0.6, 0.7$. To compare these imbalanced settings with a balanced setting, we also conduct experiments with graphs having completely balanced classes. For the intuitive explanation, we give the visualization showing how generated graphs look like in the supplementary material.

**Observations.** Figure 1 shows the node classification performance, i.e., f1-macro, with various class sizes. No methods consistently outperform other methods across the balanced and imbalanced settings. In the balanced setting, the very recent method GPRGNN achieves the best f1-macro due to its high expressive capability. On the other hand, interestingly the simplest algorithm SGC outperforms other complicated GNN algorithms in the imbalanced settings, $\alpha = 0.4, 0.5, 0.6, 0.7$. A few studies [26, 31, 41] have addressed a class imbalance problem with GNNs. However, they focus on typical homophilic graphs and ignore more complicated settings such as the combinations of imbalanced classes, heterophily property, and large-scale graphs. For example, widely used heterophilic datasets Texas, Wisconsin, and Cornell have imbalanced classes, i.e., their largest classes contain $55\%, 55\%, 47\%$ of nodes, respectively, while they all have five classes.

### 6.1.2 Various Edge Connection Proportions between Classes

To clarify in detail how effectively GNNs perform on graphs with various edge connection proportions between classes, we conduct experiments with fine-grained patterns of synthetic graphs in terms of edge connection proportions.

**Detailed setup.** To generate graphs with various edge connection proportions, we configure the class preference means $M^{\text{conf}}$ as follows:

$$M_{l_1 l_2}^{\text{conf}} = \begin{cases} \max(M_{l_1 l_2}^{\text{Cora}} - 0.1 * \beta, 0) & (l_1 = l_2) \\ M_{l_1 l_2}^{\text{Cora}} + 0.1 * \beta/(k-1) & (l_1 \neq l_2) \end{cases}, \tag{4}$$

where $\beta$ is a parameter controlling the homophily/heterophily property in a graph and $M^{\text{Cora}}$ indicates the class preference mean of Cora (Eq. (1)). Note that since the average diagonal elements of $M^{\text{Cora}}$ is 0.81, synthetic graphs without modifying the configuration generated from the Cora dataset are also

homophilic graphs. If $\beta$ is small, classes have many intra-edges, i.e., homophily property. In contrast, if $\beta$ is large, classes have few edges internally, i.e., heterophily property. For intuitive explanation, we show how generated graphs look like when varying $\beta$ in the supplementary material. We generate synthetic graphs for each $\beta \in [0, 2, 4, 6, 8]$.

**Observations.** Figure 2 shows the node classification performance with various edge connection proportions, i.e., class preference means. First, all models work well in the homophily setting (see the rightmost points). On the other hand, in the heterophily setting (see the leftmost points), GNNs generalizing to heterophilic graphs perform well such as H2GCN, FSGNN, and GPRGNN, while typical GNNs such as GCN and GAT fail to perform well. However, they obtain the marginal improvement from MLP in the heterophily setting despite their complicated designs. This indicates that GNNs generalizing heterophilic graphs hardly utilize the topology information and almost ignore the information in this setting even if graphs have the strong heterophily property. Note that the average diagonal elements of $\boldsymbol{M}^{\mathrm{conf}}$ is 0.03 when $\beta = 8$, i.e., there are only few edges inside a class. Since GraphSAGE adopts ego- and neighbor-embedding separation, it obtains relatively high f1-macro scores in the heterophily setting. The very recent method LINKX cannot achieve the state-of-the-art f1-macro scores. This is because we just follow the hyperparameter search space specified in [18] (for details, see the hyperparameter search space described in the supplementary material). Broader search space may increase the classification performance. Finally, all models obtain low f1-macro scores when edges are almost randomly generated between classes (see plots locating around $\beta = 6$ in the horizontal axis). This is because the values of class preference means are similar to uniform values in this case[7].

### 6.1.3 Various Attribute Values

We aim to clarify how largely attribute values affect the performance of GNNs by using fine-grained patterns of synthetic graphs in terms of attribute values.

**Detailed setup.** We mix the attribute-class correlation $\boldsymbol{H}^{\mathrm{Cora}}$ of Cora (Eq. (2)) into a uniform distribution with a certain degree $\gamma$, to generate graphs with various attribute-class correlations, i.e., from biased attributes for classes to random attributes. The mixing calculation formula to configure the attribute-class correlations $\boldsymbol{H}^{\mathrm{conf}}$ is as follows: $\boldsymbol{H}^{\mathrm{conf}} = (\boldsymbol{H}^{\mathrm{Cora}} + \gamma c)/(1 + \gamma)$, where $c$ is the average value of $\boldsymbol{H}^{\mathrm{Cora}}$, i.e., $c = \sum_{i=1}^{d} \sum_{j=1}^{k} \boldsymbol{H}_{ij}^{\mathrm{Cora}}/(d * k)$. If $\gamma = 0$, $\boldsymbol{H}^{\mathrm{conf}}$ corresponds to the attribute-class correlation of the original. If $\gamma$ is large, $H^{\mathrm{conf}}$ is close to uniform, i.e., $H^{\mathrm{conf}}$ is less informative to predict node labels. We generate graphs with configured attribute-class correlations for $\gamma \in [16, 4, 1, 0]$. To compare the above setting with random attributes, we also conduct experiments with graphs having uniform attribute-class correlation, i.e., every element of $\boldsymbol{H}^{\mathrm{conf}}$ is $c$.

**Observations.** Figure 3 shows the node classification performance with various attribute values. We observe that most models obtain low f1-macro scores when attribute values are close to random, i.e., $\gamma$ is large. The reason that SGC works well in some cases is that it does not over-fit large classes due to the simplicity of its model. Remember that Cora has weakly imbalanced classes as we discussed in Section 6.1.1. However, SGC is not stable across various settings. Another observation is that the very recent method GPRGNN consistently performs well compared with other GNNs. Finally, MLP achieves the larger performance gain than most GNNs between the leftmost and rightmost points (random and biased attributes) in Figure 3. This indicates some overlap between the contributions of the topology and attributes to node classification.

### 6.1.4 Various graph sizes

To clarify how graph sizes affect the performance of GNNs, we conduct experiments with graphs having various sizes and the same characteristics other than sizes.

**Detailed setup.** To generate graphs with various sizes, we set pairs of the numbers of nodes and edges $(n, m)$ to $[(3000, 5000), (6000, 10000), (9000, 15000), (12000, 20000), (15000, 25000)]$. For the sake of simple comparison, we multiply the smallest graph size $(3000, 5000)$ by constant factors $(2, 3, 4, 5)$ to obtain larger graphs.

---

[7]Note that attributes also are almost random in this case since the class structure is shared by the edge and attribute generation steps of GenCAT.

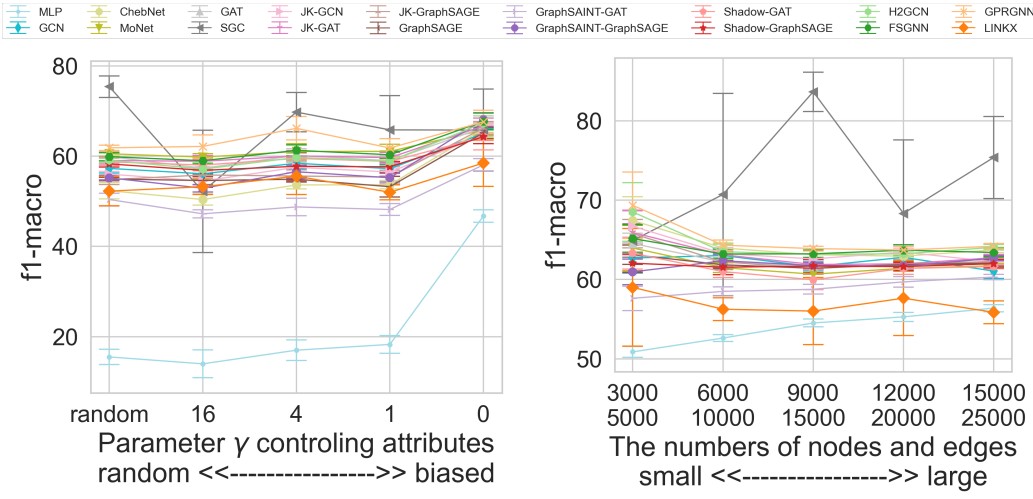

Figure 3: Classification performance on graphs with various attribute values.

Figure 4: Classification performance on graphs with various graph sizes.

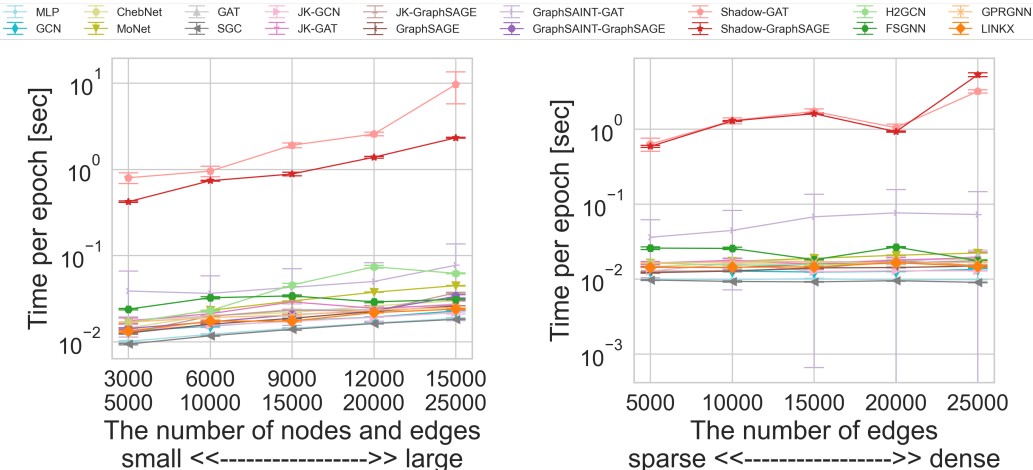

Figure 5: Training time per epoch of GNNs on graphs with various sizes.

Figure 6: Training time per epoch of GNNs on graphs with various numbers of edges.

**Observations.** Figure 4 shows the classification performance with various graph sizes. We observe that several GNNs obtain lower f1-macro scores when graphs are larger. GNNs may tend to over-fit larger classes and give less priority to smaller classes when larger graphs can provide more patterns of their subgraphs for model training, because the loss functions of most GNNs are designed to increase their accuracy. Actually, when we use accuracy as a metric, most GNNs achieve better scores as graphs grow (we show the results regarding accuracy in the supplementary materials). As the same reason in Section 6.1.3, SGC works well in some cases but is not stable across various graph sizes. MLP increases its classification performance as graph sizes are large, since it appropriately fits to node attributes by using more train data from larger graphs.

## 6.2 Training Efficiency on Synthetic Graphs with Various Graph Sizes

**Various numbers of nodes and edges.** To investigate how graph sizes, i.e., the numbers of nodes and edges, affect training efficiency, we measure the total execution time of GNNs for model training with synthetic graphs with various sizes. Note that we use the same synthetic graphs in Section 6.1.4. We tune the hyperparameters of GNNs which maximize f1-macro scores by using a grid search.

Figure 5 shows the training time per epoch of GNNs on graphs with various sizes. We observe that all methods tend to need longer training time per epoch as graphs grow, since the sizes of matrices storing the topology and attribute information increase. Shadow-GAT and Shadow-GraphSAGE require longer execution time than other GNNs because they execute a number of convolutional operations on sampled subgraphs.

**Various numbers of edges and fixed number of nodes.** To investigate how the edge densities of graphs affect training efficiency, we measure the execution time of GNNs for model training. We set the number of edges to $[5000, 10000, 15000, 20000, 25000]$ and the number of nodes to that of nodes in Cora.

Figure 6 shows the training time per epoch of GNNs on graphs with various edge densities. Most GNNs require similar training time per epoch across sparse and dense settings since the size of the adjacency matrix is the same. They perform efficiently as long as a whole graph can be stored in a GPU memory[8]. Sampling-based GNNs, GraphSAGE, GraphSAINT, and Shadow-GNN, tend to need longer training time per epoch as the edge density becomes higher, since nodes in more dense graphs have more neighbors. The reason that the training time per epoch of Shadow-GAT and Shadow-GraphSAGE decreases at graphs with 20000 edges is that smaller numbers of sampled neighbors are selected in hyperparameter tuning to maximize their f1-macro scores. Also, the reason that the training time per epoch of FSGNN decreases at graphs with 15000 and 25000 edges is that fewer hops for feature aggregation are selected than those of other settings.

# 7 Related Work

**Benchmarking GNNs.** Several studies [8, 10, 24] have addressed benchmarking GNNs. A study [8] supports a variety of graph tasks, i.e., node classification, graph classification, link prediction, and graph regression. However, since only one or two datasets are used for each task, no deep analysis for node classification is provided. GraphWorld [24] provides limited insights for node classification with GNNs because it ignores three aspects. First, it ignores recent GNNs [3, 18, 22, 46] generalizing to heterophilic graphs, which have attracted much attention from the community. Second, the study does not care about class size distributions though they largely affect classification results. Note that a class imbalance problem has been explored in the machine learning field including graph mining [26, 31, 41]. Third, GraphWorld has not explored the efficiency of GNNs, which is one of the most common concerns of machine learning methods. Another study [10] focuses on GNNs for materials chemistry. Due to the characteristics of the data, the study considers only graph regression.

**Surveys on GNNs.** Many surveys on GNNs [33, 40, 42, 44] have been conducted. They classify existing GNNs into several categories, e.g., recurrent graph neural networks, convolutional graph networks, graph autoencoders, and spatial-temporal graph neural networks in [33]. Also, [11] provides the recent overview of GNNs. A survey [42] has addressed summarizing the current state of GNNs for graphs with the heterophily property. Since their purpose is to summarize a line of research on GNNs and discuss potential research directions, they provide no experimental results.

In summary, no work has extensively conducted experiments for node classification with GNNs in order to clarify their applicability/limitations by using graphs with various characteristics.

# 8 Conclusion, Open Questions, and Limitations

In this paper, we first present a general summary of GNNs. Second, we list real-world benchmark datasets with the homophily and heterophily properties and discuss their limitations for deep analysis on GNNs, which existing studies have typically ignored. Third, we briefly summarize existing synthetic graph generators that allow us to generate graphs suitable to evaluate GNNs. Then thanks to a variety of synthetic graphs with controlled characteristics, our empirical study reveals several interesting findings of GNNs that may be helpful for developing future algorithms, while limited benchmark datasets cannot provide such fine-grained analysis. We also provide an open-sourced PyTorch-based library to foster future research on GNNs.

---

[8]We briefly discuss the application of GNNs to large-scale graphs that exceed the size of a GPU memory in Section 8

**Open Questions.** Through our empirical studies, we figured out several open questions to developing new GNNs for node classification.

*Class imbalance.* We demonstrated that a class imbalance problem deteriorates the classification performance of most GNNs. This is because their loss functions are not designed to maximize classification performance in a class imbalance setting. Though a few studies [26, 31, 41] have addressed a class imbalance problem on a node classification task with GNNs, their main focus is homophilic and small-scale graphs. Hence, it is still an open question how to develop GNNs that work well in various and complicated settings such as the combinations of class imbalance, heterophily property, and large-scale graphs.

*Heterophily setting.* Recent GNNs [3, 22, 46] have been proposed to support homophilic and heterophilic graphs. However, the performance of MLP on the strong heterophily setting is comparable to those of such GNNs (see the leftmost points in Figure 2). This indicates that they almost ignore the topology information from heterophilic graphs and there is room for performance improvement in the heterophily setting. Hence, it is an open question how to develop GNNs that can capture the class structure from heterophilic graphs while achieving the state-of-the-art performance on homophilic graphs.

We hope our experimental results motivate researchers to develop new GNN algorithms that address the above open questions, since they are crucial to apply GNN algorithms to real applications.

**Limitations.** Due to limited space, some limitations of our work need to be acknowledged. In this study, we focus on only a single task, node classification, though it is one of the hottest tasks of GNNs. By focusing on node classification, we can provide deep analysis that clarifies the pros/cons of GNNs. Also, this study does not support edge directions, edge labels, and time-series. Several recent studies [15, 20, 37] focus on developing GNNs that work on heterogeneous graphs in which objects of different types interact with each other in various ways. It would be interesting to conduct experiments with various heterogeneous graph datasets and benchmark such GNNs.

Recently, several GNN algorithms [9, 38] have been proposed to improve the scalability of GNNs. Our empirical studies focus on a traditional setting in which a whole graph and node attributes can be stored in a GPU memory at the same time. It is our future work to address benchmarking scalable GNNs in terms of efficiency and effectiveness on large-scale graphs.

## Acknowledgments and Disclosure of Funding

This work was supported by JSPS KAKENHI Grant Numbers JP20H00583 and JST PRESTO Grant Number JPMJPR21C5.

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
