# A Statistics of Benchmark Datasets

Table 1 shows the statistics of benchmark datasets we discussed in Section 4. The datasets have diverse numbers of nodes, edges, attributes, and classes.

Table 1: Summary of benchmark datasets.

| Dataset | Nodes | Edges | Attributes | Classes |
|---|---|---|---|---|
| Cora | $2,708$ | $5,278$ | $1,433$ | 7 |
| Pubmed | $19,717$ | $44,324$ | 500 | 3 |
| Citeseer | $3,327$ | $4,552$ | $3,703$ | 6 |
| Computers | 13,752 | 245,861 | 767 | 10 |
| Photo | 7,650 | 119,081 | 745 | 8 |
| Reddit | 232,965 | 11,606,919 | 602 | 41 |
| ogbn-arxiv | $169,343$ | $1,166,243$ | 128 | 40 |
| ogbn-papers100M | $111,059,956$ | $1,615,685,872$ | 128 | 172 |
| ogbn-product | $2,449,029$ | $61,859,140$ | 100 | 47 |
| Texas | 183 | 295 | $1,703$ | 5 |
| Wisconsin | 251 | 466 | $1,703$ | 5 |
| Cornell | 183 | 280 | $1,703$ | 5 |
| Actor | $7,600$ | $26,752$ | 932 | 5 |
| Chameleon | $2,277$ | $31,421$ | $2,325$ | 5 |
| Squirrel | $5,200$ | $198,493$ | $2,089$ | 5 |
| Penn94 | 41,554 | 1,362,229 | 5 | 2 |
| Pokec | 1,632,803 | 30,622,564 | 65 | 2 |
| genius | 421,961 | 984,979 | 12 | 2 |
| Deezer-Europe | 28,281 | 97,752 | 31,241 | 2 |
| Twitch-Gamers | 168,114 | 6,797,557 | 7 | 2 |
| arXiv-year | 169,343 | 1,166,243 | 128 | 5 |
| snap-patents | 2,923,922 | 13,975,788 | 269 | 5 |
| Wiki | 1,925,342 | 303,434,860 | 600 | 5 |
| YelpChi | 45,954 | 3,846,979 | 32 | 2 |

# B Detailed Experimental Settings

## B.1 Hyperparameter Search Space

We select hyperparameters for search space of GNNs according to their papers or codebases. We show hyperparameter search space for each GNN and MLP as follows.

**MLP.**

- Weight decay: $[0, 5e\text{-}6, 5e\text{-}5, 5e\text{-}4]$
- Learning rate: $[0.002, 0.01, 0.05]$
- Early stopping: $[40, 100]$
- Hidden layer: $[64]$
- Dropout: $[0.5]$

**GCN.**

- Weight decay: $[0, 5e\text{-}6, 5e\text{-}5, 5e\text{-}4]$
- Learning rate: $[0.002, 0.01, 0.05]$
- Early stopping: $[40, 100]$
- Hidden layer: $[16, 32, 64]$
- Dropout: $[0.5]$

**Monet.**

- Weight decay: $[5e\text{-}5, 5e\text{-}4, 1e\text{-}4]$
- Learning rate: $[1e\text{-}4, 0.002, 0.001, 0.01]$
- Early stopping: $[100]$
- Hidden layer: $[64]$
- Aggregation: [mean, max]
- Dropout: $[0.5]$

**ChebNet.**

- Weight decay: $[0, 5e\text{-}6, 5e\text{-}5, 5e\text{-}4]$
- Learning rate: $[0.002, 0.01, 0.05]$
- Early stopping: $[40, 100]$
- Hidden layer: $[16, 32, 64]$
- Dropout: $[0.5]$

**GAT.**

- Weight decay: $[0, 5e\text{-}6, 5e\text{-}5, 5e\text{-}4]$
- Learning rate: $[0.002, 0.01, 0.05]$
- Early stopping: $[40, 100]$
- Hidden layer: $[16, 32, 64]$
- Dropout: $[0.5]$
- Heads: $[8]$
- Output heads: $[1, 4, 8]$

**GraphSAGE.**

- Weight decay: $[0, 5e\text{-}6, 5e\text{-}5, 5e\text{-}4]$
- Learning rate: $[0.002, 0.1, 0.7]$
- Epochs: $[500]$
- Early stopping: $[40, 100]$
- Hidden layer: $[64, 128]$
- Layers: $[2]$
- Dropout: $[0.5]$

**SGC.**

- Weight decay: $[0, 5e\text{-}6, 5e\text{-}5, 5e\text{-}4]$
- Learning rate: $[0.2]$
- Epochs: $[100]$
- Early stopping: $[40]$

**JKNet.**

- Skip connection: [cat, max, lstm]

As for the hyperparameters of base models, GCN, GAT, and GraphSAGE, we follow the best hyperparameters of base models.

**GraphSAINT (-GAT and -GraphSAGE).**

- Weight decay: $[0]$
- Learning rate: $[0.002, 0.01, 0.01]$
- Epochs: $[300]$
- Early stopping: $[40]$
- Hidden layer: $[128, 256]$
- Dropout: $[0, 0.1, 0.2]$
- Heads: $[8]$
- Output heads: $[1]$
- Walk length: $[2, 4]$
- Sample coverage: $[50]$
- Root: $[1500, 2000]$

**shaDow-GNN (-GAT and -GraphSAGE).**

- Weight decay: $[0]$
- Learning rate: $[0.002, 0.01, 0.01]$
- Epochs: $[200]$
- Early stopping: $[40]$
- Hidden layer: $[128, 256]$
- Dropout: $[0, 0.1, 0.2]$
- Heads: $[8]$
- Output heads: $[1]$
- Depth: $[2]$
- Number of neighbors: $[5, 10, 20]$
- Batch size: $[64, 128]$

**FSGNN.**

- Weight decay: $[0, 5e\text{-}6, 5e\text{-}5, 5e\text{-}4]$
- Learning rate: $[0.002, 0.01, 0.05]$
- Early stopping: $[40, 100]$
- Weight decay for attention: $[0.001, 0.01, 0.1]$
- Dropout: $[0.5]$
- Hidden: $[64]$
- Number of layers: $[3, 8]$

**GPRGNN.**

- Weight decay: $[0, 5e\text{-}6, 5e\text{-}5, 5e\text{-}4]$
- Learning rate: $[0.002, 0.01, 0.05]$
- Early stopping: $[40, 100]$
- Dropout: $[0.5]$
- Parameter initializing attention: $[0.1, 0.5, 0.9]$
- Number of propagation layer: $[10]$
- Number of layer of MLP: $[3]$

**LINKX.**

- Weight decay: $[0.001]$
- Learning rate: $[0.002, 0.01, 0.05]$
- Hidden layer: $[64, 128]$
- Dropout: $[0, 0.5]$
- Number of edge layers: $[1, 2]$
- Number of node layers: $[1, 2]$
- Number of prediction layer: $[2, 3, 4]$

## B.2 Best Parameters

For the reproduction of Figures 1, 2, 3, 4, 5, and 6, we report the best set of hyperparameters for each experiment. Since we use over 20 synthetic graphs and 17 GNN models, we provide the best parameter sets in our codebase. Please see README in the repository, to find the best parameter sets.

## B.3 Source Codes of Baseline GNNs

Table 2 summarizes the URLs to download the baseline codes.

Table 2: URLs of baseline codes.

| Baseline | URLs |
|----------|------|
| SGC | https://github.com/Tiiiger/SGC |
| H2GCN | https://github.com/GemsLab/H2GCN |
| FSGNN | https://github.com/sunilkmaurya/FSGNN |
| GPRGNN | https://github.com/jianhao2016/GPRGNN |

As for GCN, MoNet, ChebNet, GAT, JKNet, GraphSAGE, GraphSAINT, Shadow-GNN, and LINKX, PYTORCH GEOMETRIC[8] provides their model architectures, so we implemented the GNN algorithms based on PYTORCH GEOMETRIC. As for H2GCN, the authors implemented it in TENSORFLOW, so we reimplemented it in PYTORCH for fair comparisons.

## C Visualization of Generated Graphs

To intuitively show how generated graphs with configured characteristics look like, we visualize several generated graphs used in our empirical studies. We first show graphs with various class size distributions in Figure 7. In Figure 7a, all classes share the same size. When we set $\alpha = 0.5$, the largest class (see the purple class in Figure 7b) includes $50\%$ of nodes in a graph. Setting $\alpha = 0.7$ is a more extreme case. Figure 7c shows a graph where the large classes (the purple, green, and blue classes) include most nodes in a graph and a few nodes belong to other classes (the red, brown, and orange classes).

Next, we show graphs with various class preference means in Figure 8, which are used in Section 6.1.2. Figure 8a shows a graph with a strong homophily property, i.e., nodes in each class are densely connected. When we set $\beta = 2$, each class has more edges connecting to other classes than $\beta = 0$ while nodes in the same class are still plotted close together in Figure 8b. In Figure 8c, the class separation becomes more ambiguous, compared with smaller $\beta$. Graphs with larger $\beta$ than $4$ look like random graphs (see Figures 8d and 8e), Also, Figure 9 depicts heatmaps representing class preference means to show the exact values used in our experiments. The class preference means in Figures 9a, 9b, 9c, 9d, 9e are used to generate graphs in Figures 8a, 8b, 8c, 8d, 8e, respectively. The larger diagonal elements of class preference means indicate more edges within a class.

Finally, we visualize the attributes of generated graphs in Figure 10. In our experiments, we generate attributes that are binary for each dimension since the Cora dataset has a binary value for each attribute dimension. However, high-dimensional datasets with binary attributes are not suitable to visualize. To address this, we utilize t-SNE[9] to reduce the attribute dimension to two-dimensional

---

[8] https://github.com/pyg-team/pytorch_geometric
[9] https://scikit-learn.org/stable/modules/generated/sklearn.manifold.TSNE.html

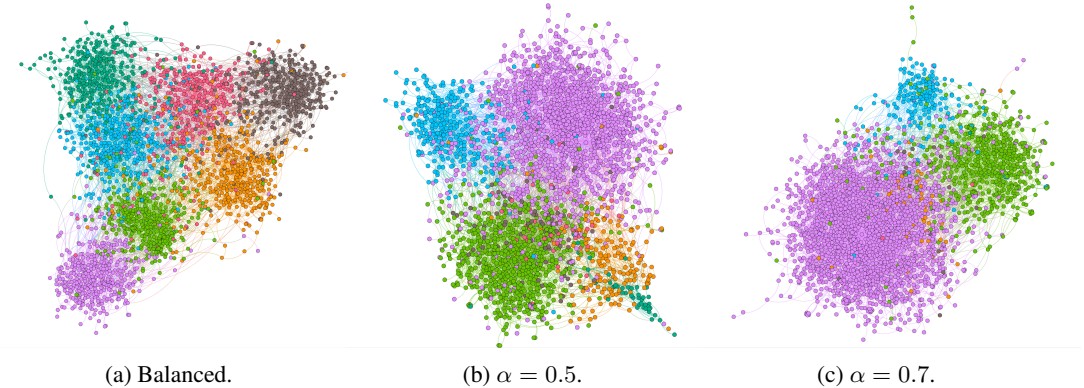

(a) Balanced.    (b) $\alpha = 0.5$.    (c) $\alpha = 0.7$.

Figure 7: Visualization showing graphs with various class size distributions. Colors indicate node class labels.

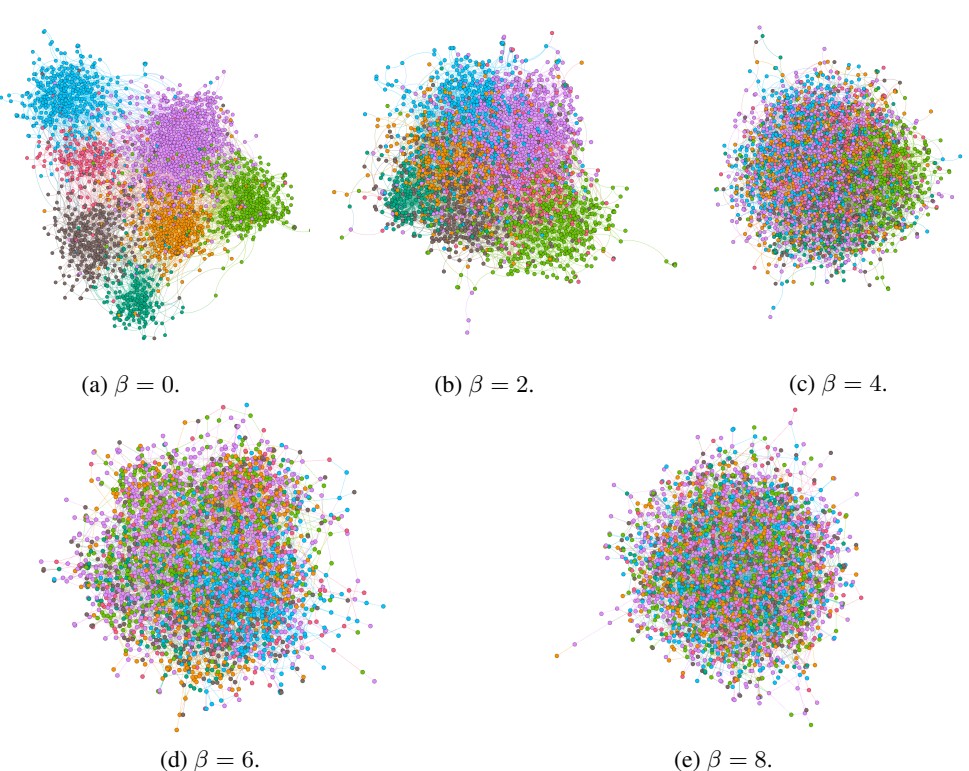

(a) $\beta = 0$.    (b) $\beta = 2$.    (c) $\beta = 4$.

(d) $\beta = 6$.    (e) $\beta = 8$.

Figure 8: Visualization showing graphs with various class preference means. Colors indicate node class labels.

real numbers. Figure 10a shows attributes generated by using the attribute-class correlation extracted from the Cora dataset, i.e., $\gamma = 0$. Nodes in the same class (nodes with the same color) are plotted close together, e.g., many nodes colored with blue are plotted in the right bottom part. On the other hand, Figure 10b depicts generated attributes when setting $\gamma = 4$. The attributes are almost randomly plotted because the configured attribute-class correlation is mixed with the average value of Cora's attribute-class correlation in this setting, i.e., the configured attribute-class correlation has a weaker bias for some particular classes.

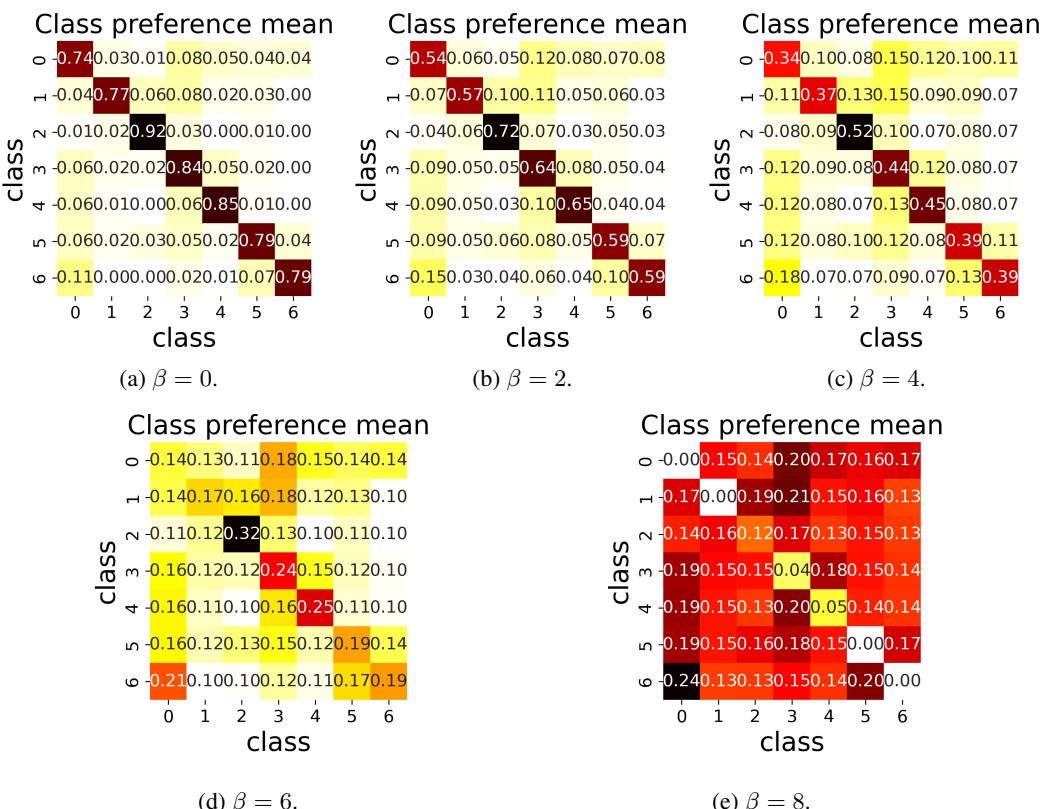

Figure 9: Class preference means used in our experiments. Each cell represents the class preference mean of the generated graph.

# D    Accuracy Analysis on Node Classification

In Section 6, we utilize f1-macro which can better reflect the performance on minority classes. We also use accuracy to evaluate classification quality since it is commonly used in existing studies developing GNNs. We use the same hyperparameter search space to Section 6 and report the best parameter set for each setting in our codebase.

## D.1    Various Class Size Distributions

Figure 11 shows accuracy on graphs with various class size distributions, which are the same setting to Section 6.1.1. We observe that all models obtain better accuracy as the size of the largest class increases. This result is the opposite of that of f1-macro. This suggests the risk of evaluating accuracy alone when class size distributions are not balanced. Another observation is that a very recent method GPRGNN achieves the highest accuracy across various class size distributions due to its model expressive capability. The reason that accuracy scores are not stable across various class size distributions is that accuracy highly depends on the characteristics of large classes, i.e., classes to which many nodes belong. Actually, we observe that the largest class has many edges within the class (i.e., the class is easy to classify) when $\alpha = 0.4, 0.6$. So, the accuracy scores in the settings are high. Note that this trend happened by chance because we shuffled class IDs before generating graphs.

## D.2    Various Edge Connection Proportions

Figure 12 shows accuracy on graphs with various edge connection proportions between classes, which is the same setting to Section 6.1.2. All GNNs perform well in the homophily setting, i.e., $\beta = 0$. On the other hand, MLP and GNNs considering the heterophily perform well when $\beta = 8$. We observe that GPRGNN achieves the highest accuracy for all patterns of $\beta$ since it can utilize the information from high-order neighbors by its deep propagation layer.

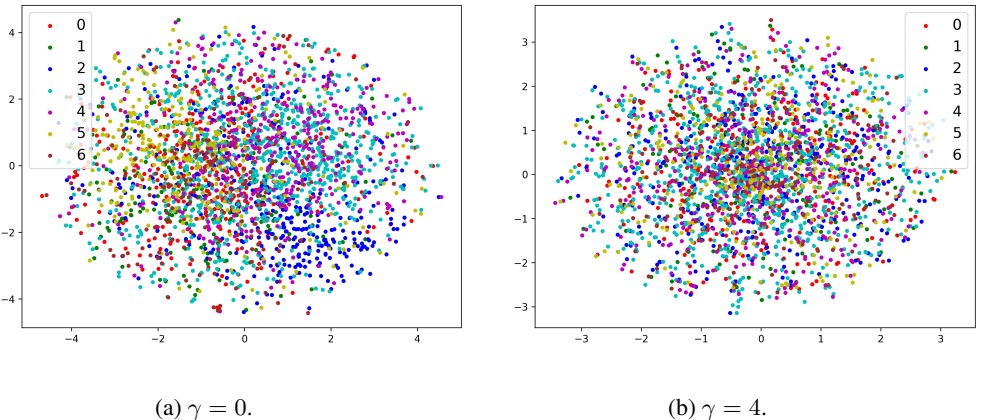

(a) $\gamma = 0$.               (b) $\gamma = 4$.

Figure 10: Two-dimensional projection of attributes. Each dot indicates a node and colors indicate node class labels.

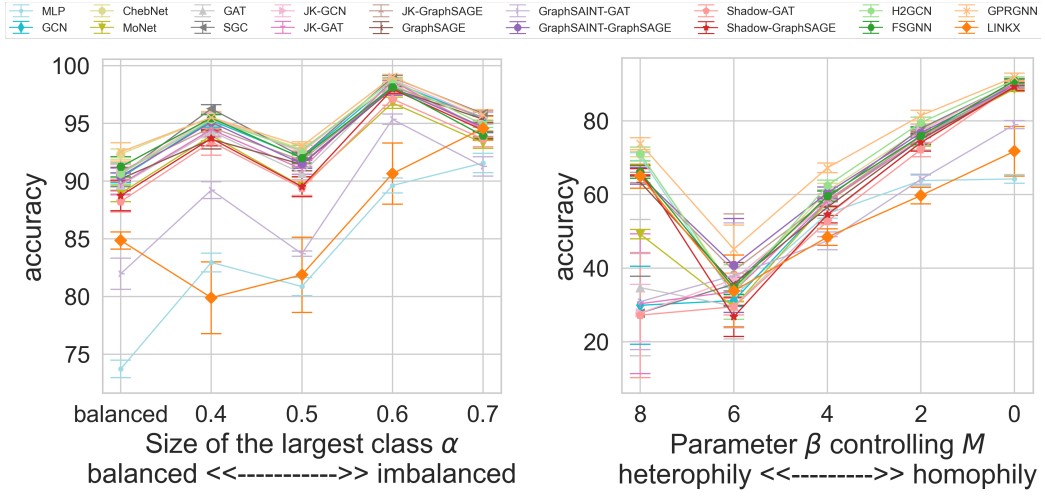

Figure 11: Accuracy on graphs with various class size distributions.

Figure 12: Accuracy on graphs with various edge connection proportions.

### D.3 Various Attribute Values

Figure 13 shows accuracy on graphs with various attribute values, which is the same setting to Section 6.1.3. We observe that all models obtain lower accuracy as attribute values are closer to random and have a weaker bias for some classes, i.e., $\gamma$ is larger. Also, GPRGNN achieves the highest accuracy across various settings.

### D.4 Various Graph Sizes

Figure 14 shows accuracy on graphs with various graph sizes, which is the same setting to Section 6.1.4. We observe that most GNNs tend to obtain higher accuracy as graphs grow. This is because the models can fit to given graphs better by using more patterns of subgraphs from larger graphs. Since generated graphs have weakly imbalanced classes in this setting, interestingly we observe a different behavior between scores of f1-macro (in Figure 4) and accuracy (in Figure 14), i.e., accuracy slightly increases as graphs grow while f1-macro tends to decrease.

Through these experiments with accuracy, we clarify that careful selection of evaluation metrics is important to fairly compare GNNs since their classification performance depends on the metrics. Practitioners need to choose a GNN algorithm that is suitable for their objectives. From the viewpoint

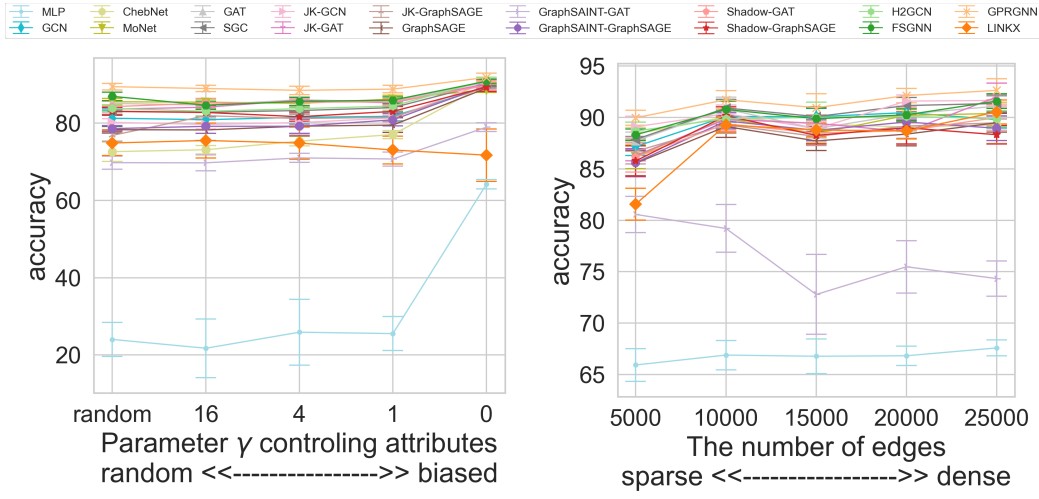

Figure 13: Accuracy on graphs with various attribute values.

Figure 14: Accuracy on graphs with various graph sizes.

of research, though a few researches [31, 41] tackled a class imbalance problem on graphs, various and complicated settings, e.g., the combinations of class imbalance, heterophily property, and large-scale graphs, are still challenging. These complicated settings need to be addressed towards real applications.

# E    Experiments for Large Datasets

We explored small datasets in our empirical studies. So, we also provide experimental results using larger datasets in Figure 15. We set pairs of the numbers of nodes and edges $(n, m)$ to $[(60000, 100000), (120000, 200000)]$. Note that a graph with 120000 nodes is a size of the same magnitude as ogbn-arxiv (169343 nodes). Due to time limitations, we use the hyperparameter tuned to graphs with 15000 nodes and 25000 edges and also plot the results of the graphs in the figure for comparison. Figure 15a shows f1-macro scores and Figure 15b shows training time per epoch.

**Observations.** First, H2GCN suffers from out-of-memory on large graphs since it computes a high-order adjacency matrix, i.e., an exact 2-hop away matrix, which requires the power of the adjacency matrix. Then, the results of other most GNNs on large graphs have similar tendencies to those on small graphs.

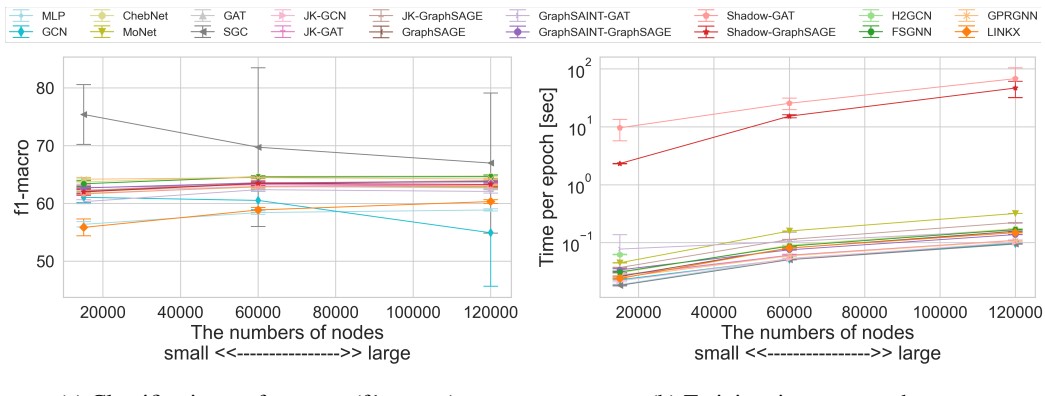

(a) Classification performance (f1-macro).

(b) Training time per epoch.

Figure 15: Experiments on large graphs.