# OpenReview forum: "Beyond Real-world Benchmark Datasets: An Empirical Study of Node Classification with GNNs"
_NeurIPS.cc/2022/Track/Datasets_and_Benchmarks — NeurIPS 2022 Datasets and Benchmarks _

### Official Review · Reviewer_Pr2L · 2022-07-25
**A Review of Beyond Real-world Benchmark Datasets: An Empirical Study of Node Classification with GNNs**

**Rating:** 6
**Confidence:** 4
**Clarity:** This paper clearly describes the expe…

**Strengths:**

This work briefly summarized four different characteristics  and their influencies on diffenent kinds of GNNs. They discover that  by generalizing to  heterophilic graphs , the networks  may perform better  classification in a heterophily setting from a graph-agnostic
 classifier MLP.

**Weaknesses:**

A variety methods has been adopted to solve the imbalance problems in positive and negative classes, such as negative sampling, and has achieved great performance in node classifiaction[1] . It maybe more appropriate and fair to adopt negativate sampling while making comparisons between different GNNs.



[1] Zhen Yang, Ming Ding, Chang Zhou, Hongxia Yang, Jingren Zhou, and Jie Tang. 2020. Understanding Negative Sampling in Graph Representation Learning. In Proceedings of the 26th ACM SIGKDD International Conference on Knowledge Discovery & Data Mining (KDD '20). Association for Computing Machinery, New York, NY, USA, 1666–1676. https://doi.org/10.1145/3394486.3403218

**Additional Feedback:**

N/A

**Correctness:**

According to the author, there are two main claims made in the paper:
"GNNs including the state-of-the-art methods suffer from a class imbalance problem that typically deteriorates the performance of multi-class classification. Interestingly, the simplest algorithm SGC outperforms recent complicated GNN algorithms in a class imbalanced setting because the complicated algorithms tend to over-fit to major classes."
"GNNs generalizing to graphs with few edges within each class (we call heterophilic graphs),provide marginal classification performance gains in a heterophily setting from a graph-agnostic classifier MLP. This indicates that such GNNs almost ignore the topology information in the setting."

The above are plausible proposals and have some experiments with finetuned parameters as backups.


**Documentation:**

There is sufficient detail for reproduction

**Relation To Prior Work:**

This work clearly discusses the differences between previous work

**Summary And Contributions:**

To analyze the impact of different chracterisics of graphs on  the strengths and weaknesses of real word GNNs,this paper proposes to use a synthetic graph generator to generate graphs with specified features. They mainly explored from four aspects:1. balanced or imbalanced class size distributions 2.homophilic or. heterophilicedge connection proportions between classes 3. biased or. random attribute values and small or large graph sizes

---

> ### Author Response · Authors · 2022-08-12
> **Response**
>
> Dear Reviewer Pr2L,
>
> We really appreciate your support for our work. Thank you for agreeing with our proposals and insights.
> We hope that our responses can address your concern below:
>
> > A variety methods has been adopted to solve the imbalance problems in positive and negative classes, such as negative sampling, and has achieved great performance in node classiaction [1] . It maybe more appropriate and fair to adopt negativate sampling while making comparisons between dierent GNNs.
>
> In our understanding, the paper [1] does not address a class imbalance problem. In the paper, positive pairs indicate node pairs where there are edges between them and negative pairs indicate node pairs that are not directly connected. The focus of the paper is on how to properly choose negative pairs from a number of candidates since graphs are typically sparse.
> This is an imbalance problem of positive pairs vs. negative pairs, and not an imbalance problem regarding node class labels, i.e., major classes vs. minor classes, which we pointed out in our paper. As we have already described in our paper, a few existing works [2, 3] have addressed a class imbalance problem in line with a node classification task with graph neural networks.
> Note that GNNs used in our paper do not utilize edge negative sampling since their models are trained by using nodes with class labels, i.e., supervised training. In contrast, considering a node representation learning task, it would be interesting to explore how negative sampling approaches affect downstream node classification results by using various synthetic datasets.
>
> Your comments look more positive than your actual rating (6: Marginally above acceptance threshold). If your concern has been fully addressed, we hope you could raise your rating.
>
> **References**
>
> [1] Zhen Yang, Ming Ding, Chang Zhou, Hongxia Yang, Jingren Zhou, and Jie Tang. Understanding Negative Sampling in Graph Representation Learning. In KDD, 2020.
>
> [2] Liang Qu, Huaisheng Zhu, Ruiqi Zheng, Yuhui Shi, and Hongzhi Yin. Imgagn: Imbalanced network embedding via generative adversarial graph networks. In KDD, 2021.
>
> [3] Zheng Wang, Xiaojun Ye, Chaokun Wang, Jian Cui, and S Yu Philip. Network embedding with completely-imbalanced labels. IEEE TKDE, 2020.

---

### Official Review · Reviewer_T9EA · 2022-07-25

**Rating:** 8
**Confidence:** 4
**Clarity:** Yes, I think the paper is well written.

**Strengths:**

This paper is well motivated. As the authors point out, the existing evaluation of GNNs lacks fine-grained analysis from various characteristics of graphs.  I applaud the authors' efforts in experiments. To clearly and comprehensively clarify the strengths and weaknesses of GNNs, they conduct quite enormous experiments including 16 GNNs with different features. The authors also use nearly every dataset commonly used in the node classification task. This paper has many contributions from both technical and conceptual perspectives. I applaud the authors' efforts to clearly divide the characteristics of real-world graphs into four parts and point out how to manually control the characteristics in details.




**Weaknesses:**

As the authors imply, they only conduct relative experiments on node classification tasks. The GNNs can be applied to various tasks including node classification, link prediction, etc. Large datasets like Reddit are not included according to the paper because of the space limit. Nowadays there are many sampling functions in PyTorch to do with large-scale datasets so experiments can be conducted with limited space. Also, as the main contribution of this paper, the figures are set quite small and indistinguishable. In Figure 1, the setting of the size is not sufficent enough compared to other figures.

**Additional Feedback:**

Anything to replenish.

**Correctness:**

Yes, I think the evaluation methods and experiment design appropriate and performed correctly.

**Documentation:**

Yes, this work mentioned available code sources to conduct reproducibility experiments. Many experiment details are also mentioned in the paper.

**Ethics:**

No, this work uses common world used datasets.

**Relation To Prior Work:**

This work mainly focuses on four characteristics of graphs

**Summary And Contributions:**

This paper points out that the existing evaluation of GNNs lacks fine-grained analysis from various characteristics of graphs. Based on this, the authors conduct extensive experiments with a synthetic graph generator on different datasets and clarify the strengths and weaknesses of GNNs from four chosen characteristics.

---

> ### Author Response · Authors · 2022-08-12
> **Response**
>
> Dear Reviewer T9EA,
>
> We really appreciate your comments and your support for our work. We hope our responses can address your concerns.
>
> > Large datasets like Reddit are not included according to the paper because of the space limit.
>
> We added the Reddit dataset into the dataset descriptions in Section 4.1. To address the concern regarding large datasets, we have additionally conducted relatively large datasets and reported the results in Figure 15 in the updated supplementary material. The summary of the experiments is two-fold: 1) the results of most GNNs on large graphs have the same tendency to those on small graphs, and 2) GNNs using a high-order adjacency matrix such as H2GCN cannot scale well.
>
> > In Figure 1, the setting of the size is not sufficient enough compared to other figures.
>
> To address this concern, we have made all figures larger so that it is easy to see and compare the results.
>
> Thank you again for applauding our paper and efforts. We hope that our revisions have appropriately solved your concerns.

---

### Official Review · Reviewer_Ft6Y · 2022-07-27
**Incorrect motivation with not enough contribution**

**Rating:** 5
**Confidence:** 3
**Correctness:** Yes
**Clarity:** Yes

**Strengths:**

I would like to leave it to further discussion.


**Weaknesses:**

1. Incorrect motivation. GNNs are designed to cope with real-world graph-related problems, which should be tested on real data. From my perspective, testing GNNs performance on the generated graph is not a good motivation.
2. The contribution is not big enough. It seems like the authors simply adopt previous research GenCat to generate graphs as the benchmark. The contribution of this paper is not big enough.
3. How do authors identify the four characteristics? The topology characteristics (e.g. density)  are not studied. Topology structure is the core difference between graph data and other kinds of data.


**Additional Feedback:**

N/A

**Documentation:**

Sufficient

**Ethics:**

There are no ethical concerns

**Relation To Prior Work:**

Yes

**Summary And Contributions:**

This paper argues using generated graphs to evaluate GNN performance fully. Making use of their previous work GenCAT as the graph generator, the authors develop a benchmark to test GNN’s performance with respect to 4 graph attributes: Class size distribution, edge connection proportions, attribute values, and graph size. From my perspective, the motivation that generating graphs to test GNN’s performance is incorrect, and the contribution of this paper is also very limited.

---

> ### Author Response · Authors · 2022-08-12
> **Response 2/2**
>
> **References**
>
> [1] Eli Chien, Jianhao Peng, Pan Li, and Olgica Milenkovic. Adaptive universal generalized pagerank graph neural network. In ICLR, 2021.
>
> [2] Palowitch J, Tsitsulin A, Mayer B, Perozzi B. GraphWorld: Fake Graphs Bring Real Insights for GNNs. In KDD, 2022.
>
> [3] Vijay Prakash Dwivedi, Chaitanya K Joshi, Thomas Laurent, Yoshua Bengio, and Xavier Bresson. Benchmarking graph neural networks. arXiv preprint arXiv:2003.00982, 2020.
>
> [4] Jiong Zhu, Yujun Yan, Lingxiao Zhao, Mark Heimann, Leman Akoglu, and Danai Koutra. Beyond homophily in graph neural networks: Current limitations and effective designs. In NeurIPS, 2020.
>
> [5] Seiji Maekawa, Yuya Sasaki, George Fletcher, and Makoto Onizuka. Gencat: Generating attributed graphs with controlled relationships between classes, attributes, and topology. arXiv preprint arXiv:2109.04639, 2021.
>
> [6] Derek Lim, Felix Hohne, Xiuyu Li, Sijia Linda Huang, Vaishnavi Gupta, Omkar Bhalerao, and Ser Nam Lim. Large scale learning on non-homophilous graphs: New benchmarks and strong simple methods. In NeurIPS, 2021.
>
> [7] Derek Lim, Xiuyu Li, Felix Hohne, and Ser-Nam Lim. New benchmarks for learning on non-homophilous graphs. In Workshop on Graph Learning Benchmarks (GLB 2021) at WWW, 2021.
>
> [8] Jiong Zhu, Ryan A Rossi, Anup Rao, Tung Mai, Nedim Lipka, Nesreen K Ahmed, and Danai Koutra. Graph neural networks with heterophily. In AAAI, 2021.
>
> [9] Zheng Wang, Xiaojun Ye, Chaokun Wang, Jian Cui, and S Yu Philip. Network embedding with completely-imbalanced labels. IEEE TKDE, 33(11), 2020.
>
> [10] Tianxiang Zhao, Xiang Zhang, and Suhang Wang. Graphsmote: Imbalanced node classification on graphs with graph neural networks. In WSDM, 2021.
>
> [11] Liang Qu, Huaisheng Zhu, Ruiqi Zheng, Yuhui Shi, and Hongzhi Yin. Imgagn: Imbalanced network embedding via generative adversarial graph networks. In KDD, 2021.

---

> ### Author Response · Authors · 2022-08-12
> **Response 1/2**
>
> Dear Reviewer Ft6Y,
>
> We really appreciate your comments. We first clarify our contributions and then provide point-to-point responses that address your concerns.
>
> First of all, we would like to clarify the contributions of our paper. We present new insights from our empirical studies using various synthetic graphs suitable for the evaluation and also publicly provide an evaluation framework for future research. Note that the webpage of NeurIPS 2022 Datasets and Benchmarks Track (https://neurips.cc/Conferences/2022/CallForDatasetsBenchmarks) says “systematic analyses of existing systems on novel datasets that yield important new insight are also in scope.”. Hence, our work is well suited to the Datasets and Benchmarks Track.
>
>
> > Incorrect motivation. GNNs are designed to cope with real-world graph-related problems, which should be tested on real data. From my perspective, testing GNNs performance on the generated graph is not a good motivation.
>
> Many existing studies evaluated the performance of GNNs by using synthetic graphs [1, 2, 3, 4], so it is common to use synthetic graphs for revealing the characteristics of GNNs.
> Actually, reviewers #EmbW and #T9EA agree on the correctness of our motivation and the importance of GNN performance evaluations using synthetic graphs. Reviewer #EmbW says “A synthetic dataset (with suitable tuning parameters) and benchmark for GNNs is a valuable contribution” and Review # T9EA says “This paper is well motivated. As the authors point out, the existing evaluation of GNNs lacks fine-grained analysis from various characteristics of graphs”. Hence, the motivation of our paper would not be incorrect.
>
> > The contribution is not big enough. It seems like the authors simply adopt previous research GenCat to generate graphs as the benchmark. The contribution of this paper is not big enough.
>
>
> The contributions of our paper do not overlap those of the GenCAT paper [5]. First, our paper focuses on empirical studies using 16 GNN methods and various synthetic graphs suitable for their evaluation, not just a graph generation problem. Also, as we clearly described in Section 1 and 8, our contributions are that we provide interesting insights and open questions for future research through our empirical studies. In contrast, the contributions of GenCAT are two-fold; 1) the work identifies graph and class features that captures the characteristics of attributed graphs with node class labels and 2) GenCAT effectively and efficiently generates graphs with controlled characteristics by using the graph and class features.
> Hence, our paper makes a new contribution, rather than simply adopting the prior research.
>
> > How do authors identify the four characteristics? The topology characteristics (e.g. density) are not studied. Topology structure is the core difference between graph data and other kinds of data.
>
> Due to time limitation and the limited pages, we focus on the four major characteristics that other works [1,6-11]  also are interested in. First, as for edge connection proportions, studies [1,6-8] including very recent researches have paid much attention with the homophily/heterophily property of graphs because the property really affects the node classification performance. Second, as for class sizes, studies [9-11] have addressed a class imbalance problem for node classification so class size distributions are important. Third, most GNNs typically use attributes of nodes so it is a natural question how much attributes contribute to the classification performance. Finally, as for graph sizes, the efficiency of methods is one of the common concerns in the machine learning field.
>
> From the above reasons, we prioritized the four major characteristics that largely affect the classification quality or training time. We agree that it would be interesting future directions to investigate how largely other aspects affect node classification quality.
>
> As for the topology characteristics, we use class preference means (i.e., edge connection proportions between classes) that flexibly capture edge density for each class (see the visualization of class preference means in Figure 9 in the updated supplementary material). Hence, for empirical studies in our paper, we do not fail to utilize the core difference between graph data and other kinds of data.
>
> We hope our responses have fully addressed your concerns and sound correct for you.

---

> ### Author Response · Authors · 2022-08-27
> **Are there any concerns we can clarify?**
>
> We would like to address your concerns since they are related to the motivation of our paper, i.e., the core part. Though we posted our responses two weeks ago, they might not fully address the concerns and weaknesses you pointed out. Could you tell me which weaknesses still remain?

---

> > ### Comment · Reviewer_Ft6Y · 2022-08-27
> > **Thanks for author's reply**
> >
> > Though some works utilize generated graphs in experiments, I still do not agree with evaluating the model's performance on generated graphs. Based on other reviewers' comments and the detailed reply, I raised my score from 3 to 5.

---

> > > ### Author Response · Authors · 2022-08-29
> > > **Thank you for your response.**
> > >
> > > Thank you for raising your score.
> > > We respect your opinion, but we believe generated graphs are useful to evaluate methods as we mentioned in our previous response. We would like to ask if there are any remaining concerns that we can clarify.

---

### Official Review · Reviewer_EmbW · 2022-07-28
**Fills an important need for evaluation of GNNs**

**Rating:** 7
**Confidence:** 3
**Correctness:** The work appears correct.

**Strengths:**

- A synthetic dataset (with suitable tuning parameters) and benchmark for GNNs is a valuable contribution.
- A good collection of results comparing an impressively large set of 16 GNN algorithms.
- The authors provide insight (based on experiments) regarding which techniques work well in different settings.
- The code appears solid, and reproduces raw data and plots found in the paper
- Observations regarding the four sets of results provide at least some insight into the strengths/weaknesses of the GNNs.
- There are some results in the supplemental giving insight into the dataset's characterisitcs.

**Weaknesses:**

Overall, I feel that this paper accomplished the goals it set. However, there are some additional aspects that could have been explored:
* More thorough analysis of the GNNs with respect to the benchmarks to better pinpoint why the GNNs do or not do well. The observations in Section 6.1 are nice, but don't provide deep insight into strengths/weaknesses. This would be difficult due to time and space limitations, since many GNNs were used.
* The benchmark is only based on characteristics from the Cora dataset. I think providing multiple benchmarks based on other datasets (like the Texas dataset) would provide a more comprehensive set of tests. To save space, the paper could just focus on analysis of the Cora dataset.



---
My original questions follow, which the authors since answered in their comments.

The choice of characteristics to study and tuning parameters seems somewhat ad-hoc. It would be useful if there was more insight into the choices. Specifically:
- Lines 144-147 say “Node classification results depend on four major characteristics of graphs having node labels…”. The paper does justify the importance of these four characteristics. But, what was the rationale for choosing these and not others? For example, why not vary class preference deviation to demonstrate core/border phenomena (see [20]). Why not vary number of classes, or degree distribution?
- Why was the Cora dataset used as a basis for the parameterizations?
- Why was (3) defined in this way, as opposed to using some other distribution?
- For the homophily/heterophily results, how was (4) derived? Why not interpolate, let’s say, between M^Cora (which displays homophily) and M^Texas (the Texas dataset which displays heterophily).

I also find it difficult to develop an intuition for the benchmark graphs. Given that these are the core of the paper, I think this is important. The reader could study the dataset themselves, but I feel that a large value of the publication would be to provide the reader more insight into the characteristics of these graphs. I realize that there is limited space, but perhaps some detail could be put in the supplemental? For example:
- What exactly does the class size distribution (3) look like for Fig. 1. Perhaps you could show a 3D graph that shows how the size of each of the seven classes varies as \alpha is varied. I am also not clear on why seven classes implies that the graph is imbalanced for \alpha = 0.5, 0.7. Isn’t it always imbalanced if it is generated according to (3)? Why were 0.5 and 0.7 chosen as endpoints?
- What exactly are the values of M^Cora? Can you somehow quantify or show the edge density for the different values of \beta?
- What do the attribute distributions actually look like E.g., Figure 6 or 7 in [20]

There are only 3 or 4 datapoints in each of the plots. Was this due to time limitations? It would be nice if there were at least a few more datapoints each.

-  It seems like understanding the generative model requires looking at [20] and/or the code. Is there a way to describe, let’s say, the attribute distributions exhibited by the benchmark (without having to add pages of detail)?

- What were the sizes of the train/validation/test sets? The supplemental lists the epochs for each algorithm, but not the size of the training set (unless I missed it).

- One concern: In [3], the cSBM model is used to generate graphs with both heterophily and homophily. Perhaps you should discuss how this compares with GenCAT?

**Additional Feedback:**

- For Section 6.1.2, would there be value in including a similar result but using a purely synthetic M matrix (not based on Cora), where \beta=0 would represent a graph with neither homophily nor heterophily (a so-called null model)? Negative values of \beta could generate heterophilic graphs, whereas positive values could generate homophilic graphs. I feel that this type of tuning parameter would be easier for researchers to use.

- I see that the datasets are available for download from Google Drive, which is great. Do you have a plan for ensuring these remain available in the long-term?


**Clarity:**

The paper is overall well written.


**Documentation:**

The code is good. It provides Python notebooks with the paper plots, and csv files of the dataset are also provided.




**Ethics:**

I have no ethical concerns.

**Relation To Prior Work:**

The summary of prior work is good.




**Summary And Contributions:**

The authors provide a dataset based on the GenCAT graph generator, for the purpose of evaluating node classification by Graph Neural Networks. They vary four characteristics of the graph: class size distribution, edge connection proportions, attribute values, and graph size, and provide benchmark results using 16 GNN techniques.

I feel that this paper accomplished its goals, and makes a good contribution. The authors make good use of the space provided. For this reason, I'm giving it a rating of "accept".
However, as described in "Weaknesses", I am still unsure if the benchmark is extensive enough for inclusion in the track.

---

> ### Author Response · Authors · 2022-08-12
> **Response 3/3**
>
> > For Section 6.1.2, would there be value in including a similar result but using a purely synthetic M matrix (not based on Cora), where \beta=0 would represent a graph with neither homophily nor heterophily (a so-called null model)?
>
> In Figure 2, \beta=6 represents graphs with neither homophily nor heterophily. In the setting, no methods can provide high f1-macro scores. For a better understanding of the configured class preference mean, we provide heatmaps representing class preference means in Figure 9 in the updated supplementary material.
>
> > I see that the datasets are available for download from Google Drive, which is great. Do you have a plan for ensuring these remain available in the long-term?
>
> All datasets used in our empirical studies can be generated from the scripts described in our codebase (see “Dataset Generation” in README). So, users can generate the same datasets by themselves. We believe that the sharing link is rather optional and is not an essential problem. Furthermore, to address your concern, we plan to upload the datasets onto our laboratory’s server and provide the link in our github repository.
>
>
>
> Thank you again for your careful and detailed reviews. We hope the above responses have fully addressed your concerns and questions.
>
> **References**
>
> [1] Derek Lim, Felix Hohne, Xiuyu Li, Sijia Linda Huang, Vaishnavi Gupta, Omkar Bhalerao, and Ser Nam Lim. Large scale learning on non-homophilous graphs: New benchmarks and strong simple methods. In NeurIPS, 2021.
>
> [2] Derek Lim, Xiuyu Li, Felix Hohne, and Ser-Nam Lim. New benchmarks for learning on non-homophilous graphs. In Workshop on Graph Learning Benchmarks (GLB 2021) at WWW, 2021.
>
> [3] Eli Chien, Jianhao Peng, Pan Li, and Olgica Milenkovic. Adaptive universal generalized pagerank graph neural network. In ICLR, 2021.
>
> [4] Jiong Zhu, Ryan A Rossi, Anup Rao, Tung Mai, Nedim Lipka, Nesreen K Ahmed, and Danai Koutra. Graph neural networks with heterophily. In AAAI, 2021.
>
> [5] Jiong Zhu, Yujun Yan, Lingxiao Zhao, Mark Heimann, Leman Akoglu, and Danai Koutra. Beyond homophily in graph neural networks: Current limitations and effective designs. In NeurIPS, 2020.
>
> [6] Zheng Wang, Xiaojun Ye, Chaokun Wang, Jian Cui, and S Yu Philip. Network embedding with completely-imbalanced labels. IEEE TKDE, 33(11), 2020.
>
> [7] Tianxiang Zhao, Xiang Zhang, and Suhang Wang. Graphsmote: Imbalanced node classification on graphs with graph neural networks. In WSDM, 2021.
>
> [8] Liang Qu, Huaisheng Zhu, Ruiqi Zheng, Yuhui Shi, and Hongzhi Yin. Imgagn: Imbalanced network embedding via generative adversarial graph networks. In KDD, 2021.
>
> [9] Yash Deshpande, Subhabrata Sen, Andrea Montanari, and Elchanan Mossel. Contextual stochastic block models. In NeurIPS, 2018.

---

> > ### Comment · Reviewer_EmbW · 2022-08-26
> > **Thank you for the detailed response (Part 2/2)**
> >
> > > All datasets used in our empirical studies can be generated from the scripts described in our codebase (see “Dataset Generation” in README). So, users can generate the same datasets by themselves. We believe that the sharing link is rather optional and is not an essential problem. Furthermore, to address your concern, we plan to upload the datasets onto our laboratory’s server and provide the link in our github repository.
> >
> > I agree that the code should provide sufficient reproducibility. However, I tried running the code (sorry I did not try earlier), and had some issues:
> >    A) Can you specify which version of Python you used in the README?
> >
> >    B) I installed the requirements in requirements.txt. However, it seems I needed to install "torch_sparse" and "torch_scatter" before I could run the code.  Should these be in requirements.txt? Then, when I ran the code, I received an error as below. Do you know if this is a configuration issue on my computer?
> > ```
> > python scripts/run_gencat.py --dataset cora
> >
> > Traceback (most recent call last):
> >   File "scripts/run_gencat.py", line 219, in <module>
> >     main(args)
> >   File "scripts/run_gencat.py", line 27, in main
> >     from models.dataset_utils import DataLoader
> >   File ".\models\dataset_utils.py", line 12, in <module>
> >     from torch_sparse import coalesce, SparseTensor
> >   File "C:\...\lib\site-packages\torch_sparse\__init__.py", line 41, in <module>
> >     from .tensor import SparseTensor  # noqa
> >   File "C:\...\lib\site-packages\torch_sparse\tensor.py", line 13, in <module>
> >     class SparseTensor(object):
> >   File "C:\...\lib\site-packages\torch\jit\_script.py", line 1128, in script
> >     _compile_and_register_class(obj, _rcb, qualified_name)
> >   File "C:\...\lib\site-packages\torch\jit\_script.py", line 138, in _compile_and_register_class
> >     script_class = torch._C._jit_script_class_compile(qualified_name, ast, defaults, rcb)
> > RuntimeError:
> > object has no attribute sparse_csr_tensor:
> >   File "C:\Users\mikem\.conda\envs\gnn\lib\site-packages\torch_sparse\tensor.py", line 511
> >             value = torch.ones(self.nnz(), dtype=dtype, device=self.device())
> >
> >         return torch.sparse_csr_tensor(rowptr, col, value, self.sizes())
> >                ~~~~~~~~~~~~~~~~~~~~~~~ <--- HERE
> > ```
> >
> > I have upgraded my rating to "accept". Overall, I feel that this paper accomplished the goals it set. However, there are some additional aspects that could have been explored (see below), which would have made it stronger.
> >
> > * More thorough analysis of the GNNs with respect to the benchmarks to better pinpoint why the GNNs do or not do well. The observations in Section 6.1 are nice, but don't provide deep insight into strengths/weaknesses. This would be difficult due to time and space limitations, since many GNNs were used.
> > * The benchmark is only based on characteristics from the Cora dataset. I think providing multiple benchmarks based on other datasets (like the Texas dataset) would provide a more comprehensive set of tests. To save space, the paper could just focus on analysis of the Cora dataset.

---

> > ### Comment · Reviewer_EmbW · 2022-08-26
> > **Thank you for the detailed response (Part 1/2)**
> >
> > I appreciate the detailed response and the addition of the new Figures in the supplemental. I am overall satisfied with the responses. I've included some more detailed responses below.
> >
> > > We agree that varying other characteristics is also interesting. However, due to time limitation and the limited pages, we focus on the four major characteristics that other works [1-8] also are interested in
> >
> > Although I would like to see additional characteristics studied, I realize that page limitations due make this difficult. I think there is good justification to study these aspects.
> >
> > > It is not obvious how to interpolate two class preference means because different datasets may have different numbers of classes. Actually, Cora has seven classes and Texas has five classes.
> >
> > This makes sense. I realize it is beyond the scope of this work (which docuses on the Cora dataset), but it would be nice to have a benchmark that allowed other numbers of classes besides just 7, and possibly including multiple benchmarks with characterisitcs from other datasets as well.
> >
> >
> > >...we added more data points so that each figure has at least five data points.
> >
> > Thank you, this helps. It appears more datapoints could also be generated by running the code.
> >
> >
> > > As you pointed out, cSBM can generate graphs with both heterophily and homophily. However, it focuses on a simple case where there are only two clusters, i.e., two classes (see the first sentence of Section 2 in [9])
> >
> > This makes sense - I agree that two classes is not enough.
> >
> > > See “Experimental Setup” of Section 6, where we have already described the sizes of the train/validation/test sets in the initial submission.
> >
> > Can you point me to the code where this split is made?
> >
> > ---
> > Some comments/questions regarding the figures:
> >
> > 1) Figures 7 and 8 - These are nice and give a better idea of the class sizes and topology.
> > 2) Figure 9 - Can you show a panel for beta=6 and beta=8? I believe you intended to do this based on line 246 of the main paper.
> > 3) Figure 10 - I like the use of t-SNE and the analysis, and I do see in Figure 10(a) that the dark blue nodes can be seen predominantly in the right bottom part. That being said, I still do find it difficult to interpret these plots.
> >
> > Would it be possible to include these figures in a Jupyter notebook in the code so that researchers could explore these features on their own?
> > I did not see them in the code.

---

> ### Author Response · Authors · 2022-08-12
> **Response 2/3**
>
> > What exactly does the class size distribution (3) look like for Fig. 1. Perhaps you could show a 3D graph that shows how the size of each of the seven classes varies as \alpha is varied. I am also not clear on why seven classes implies that the graph is imbalanced for \alpha = 0.5, 0.7. Isn’t it always imbalanced if it is generated according to (3)? Why were 0.5 and 0.7 chosen as endpoints?
>
> Thank you for the suggestion. To intuitively show how generated graphs with configured characteristics look like, we visualize generated graphs in Figure 7 in the updated supplementary material. In the figure, we show a graph with balanced classes and graphs with imbalance classes using \alpha=0.5, 0.7. Classes are always imbalanced when \alpha=0.5, 0.7 because the largest class (\rho_1^{conf}) includes 50% or 70% of nodes in graphs. The second largest class includes 50% or 70% of the rest of the nodes. Hence, classes have clearly different sizes in these scenarios, i.e., classes are imbalanced. As for the reason that we chose \alpha=0.5, 0.7, we aim to show cases where classes in graphs are strongly imbalanced (see Figure 7 in our supplementary material). Since we understand that three data points (balanced and \alpha=0.5, 0.7) are not sufficient, we added two more data points (\alpha=0.4, 0.6) in Figure 1. We hope that the visualization would help readers understand the characteristics of generated graphs.
>
> > What exactly are the values of M^Cora? Can you somehow quantify or show the edge density for the different values of \beta?
>
> To address this question, we show the exact values of class preference means with different values of \beta in Figure 9 in the updated supplementary material.
>
> > What do the attribute distributions actually look like E.g., Figure 6 or 7 in [20]
>
> Because the attributes of the Cora dataset are binary for each dimension, synthetic graphs used in our empirical studies also have attributes that are binary. To address your concern, we show 2D projection of generated attributes in Figure 10 in the updated supplementary material. We hope that the figure help readers understand generated attributes well.
>
> > There are only 3 or 4 datapoints in each of the plots. Was this due to time limitations? It would be nice if there were at least a few more datapoints each.
>
> We appreciate your suggestion. To address this, we added more data points so that each figure has at least five data points.
>
> > One question: on Line 236 it says “synthetic graphs without configuration are also homophilic graphs”. What does “without configuration” mean exactly?
>
> In this context, “without configuration” indicates that we directly used the class preference mean extracted from the Cora dataset to generate synthetic graphs. As you pointed out, the sentence is ambiguous. So, we replaced “without configuration” with “without modifying the configuration generated from the Cora dataset”.
>
> > One concern: In [3], the cSBM model is used to generate graphs with both heterophily and homophily. Perhaps you should discuss how this compares with GenCAT?
>
> First, we would like to clarify that the comparison of existing graph generators is not our main concern since the focus of our paper is to provide insights through empirical studies with GNNs. As you pointed out, cSBM can generate graphs with both heterophily and homophily. However, it focuses on a simple case where there are only two clusters, i.e., two classes (see the first sentence of Section 2 in [9]). Actually, a study [3] uses synthetic graphs with two classes, which are generated by cSBM. Since real-world graphs have various numbers of classes, we do not use cSBM for our empirical studies. To address your concern, we added a statement explaining the relationship between cSBM and GenCAT in Section 5.
>
> > It seems like understanding the generative model requires looking at [20] and/or the code. Is there a way to describe, let’s say, the attribute distributions exhibited by the benchmark (without having to add pages of detail)?
>
> To address this concern, we provide heatmaps of class preference means and the visualization of graphs and attributes (see Figure 7-10 in the updated supplementary material). We hope that the figures will help users easily understand the characteristics of generated graphs.
>
> > What were the sizes of the train/validation/test sets? The supplemental lists the epochs for each algorithm, but not the size of the training set (unless I missed it).
>
> See “Experimental Setup” of Section 6, where we have already described the sizes of the train/validation/test sets in the initial submission.

---

> ### Author Response · Authors · 2022-08-12
> **Response 1/3**
>
> Dear Reviewer EmbW,
>
> We appreciate your constructive comments regarding our paper. We hope our responses can address your concerns. Please find our detailed responses below.
>
> > Lines 144-147 say “Node classification results depend on four major characteristics of graphs having node labels...”. The paper does justify the importance of these four characteristics. But, what was the rationale for choosing these and not others? For example, why not vary class preference deviation to demonstrate core/border phenomena (see [20]). Why not vary number of classes, or degree distribution?
>
> We agree that varying other characteristics is also interesting. However, due to time limitation and the limited pages, we focus on the four major characteristics that other works [1-8] also are interested in. First, as for edge connection proportions, studies [1-5] including very recent researches have paid much attention to the homophily/heterophily property of graphs because the property really affects the node classification performance. Second, as for class sizes, studies [6-8] have addressed a class imbalance problem for node classification since class size distributions are also crucial for classification performance. Third, most GNNs typically use attributes of nodes so it is a natural question how much attributes contribute to the classification performance. Finally, as for graph sizes, the efficiency of methods is one of the common concerns in the machine learning field. We agree that it would be interesting future directions to investigate how class preference deviation and degree distributions affect node classification quality. However, from the above reasons, we prioritized the four major characteristics that largely affect the classification quality or training time. As for the number of classes, reducing the number of classes seems to be possible in the real-world graph. This is also interesting to investigate.
>
> > Why was the Cora dataset used as a basis for the parameterizations?
>
> Since the Cora dataset is one of the most widely used datasets, we believe that the target audience of our paper can easily understand the statistics of generated graphs. Adding a new basis dataset for synthetic graphs is our future work.
>
> > Why was (3) defined in this way, as opposed to using some other distribution?
>
> There are other possible ways to define class size distributions. The reason that we use Eq. (3) is that it is easy to control the size of the largest class (see Figure 7 in the updated supplementary material). We also provide detailed discussion about class sizes in the next response.
>
> > For the homophily/heterophily results, how was (4) derived? Why not interpolate, let’s say, between M^Cora (which displays homophily) and M^Texas (the Texas dataset which displays heterophily).
>
> First, a class preference mean, $M \in \mathbb{R}^{k\times k}$, represents edge connection proportions between classes, where $k$ is the number of classes. The motivation of Eq. (4) is controlling the diagonal elements of a class preference mean, i.e., the ratio of edges within classes, which many existing works [1-5] are interested in. As you can see, Eq. (4) is the simplest approach to increase/decrease the diagonal elements of a class preference mean and to decrease/increase other elements (see Figure 9 in the updated supplementary material). In this sense, we can control the homophily/heterophily property in a graph by using any single basis data, e.g., the Cora dataset. Then, the reason that we do not interpolate two class preference means from different datasets is two-fold. 1) The motivation of our paper is to conduct experiments on graphs with controlled characteristics. However, the interpolation between two existing datasets limits the characteristics of generated graphs, compared with manual configuration since the generated graphs are limited in between the two datasets. 2) It is not obvious how to interpolate two class preference means because different datasets may have different numbers of classes. Actually, Cora has seven classes and Texas has five classes. It is hard to specify which classes from different datasets are interpolated. We agree that the interpolation between two different datasets is interesting. So, such complex settings would be one of our future directions.

---

### Official Review · Reviewer_jwKc · 2022-07-28
**An empirical demonstration with experimental insights instead of a framework**

**Rating:** 5
**Confidence:** 4

**Strengths:**

-The paper writing is easy to follow.
-Several limitations on the existing use of datasets for node classification and the challenges in developing fine grained analysis with those are discussed, which forms the motivation of the paper.
-The method adopted or proposed in the paper can be used to test future GNNs' abilities in the corresponding 4 characteristics as stated.
-Extensive experiments are performed on the characteristics stated by using numerous GNN models that represent the progress of the field.

**Weaknesses:**

-Some existing work in this regard, such as GraphWorld [1] or the synthetic datasets used in [2], [3], among others, already use synthetic datasets for fine grained analysis or testing some particular graph properties. In such case, the use of synthetic datasets for deeper investigation is not novel to some extent.
-The objective of using a synthetic graph generator to test specific graph characteristics which may not be readily available in real-world dataset has already been explored in [1]. For example, the case of higher or lower homophily is explored in [1]. In this regard, the contribution of this paper seems limited.
-If graph generator is somehow a contribution of the paper, it is already presented in a prior work.

Hence, even though the insights presented after the extensive experiments seem interesting, the contribution of the paper look limited and not novel, and researchers may rely on the prior generator tool to assist in their experiments. Overall, this paper seems to be an experimental demonstration, instead of a framework presented, as far as this track is concerned.

[1] Palowitch J, Tsitsulin A, Mayer B, Perozzi B. GraphWorld: Fake Graphs Bring Real Insights for GNNs.
[2] Corso G, Cavalleri L, Beaini D, Liò P, Veličković P. Principal neighbourhood aggregation for graph nets.
[3] Vignac C, Loukas A, Frossard P. Building powerful and equivariant graph neural networks with structural message-passing.

Question:
- In line 206-207: "In fact, existing studies use only accuracy to evaluate the classification quality since they do not consider class size distributions." => I understand that the metric is altered other than accuracy to account for class imbalance generally in existing works. Are there particular set of references which support this statement in line 206-207?

**Additional Feedback:**

see "Weaknesses" field.

**Clarity:**

Yes the paper is fairly well written, except that Section 3 is longer than required and may not be needed in detail in such a paper.

**Correctness:**

The evaluation methods seem correct and appropriate, such as using appropriate metrics for evaluation, among others.

**Documentation:**

Yes, the documentation seems sufficient to support reproducibility.

**Relation To Prior Work:**

Mostly covered, except for [1] which is referenced in above in "Weaknesses" field.

**Summary And Contributions:**

This paper presents an empirical study of GNNs on node classification tasks by using a synthetic graph generator with controlled characteristics. Four major dimensions of such characteristics are discussed and are used for the controlled experiments. The paper conducts extensive experiments using such a setup and presents insights on how different settings affect the performance or capabilities of a GNN model.

---

> ### Author Response · Authors · 2022-08-12
> **Response 2/2**
>
> **References**
>
> [1] Palowitch J, Tsitsulin A, Mayer B, Perozzi B. GraphWorld: Fake Graphs Bring Real Insights for GNNs.
>
> [2] Corso G, Cavalleri L, Beaini D, Liò P, Veličković P. Principal neighbourhood aggregation for graph nets.
>
> [3] Vignac C, Loukas A, Frossard P. Building powerful and equivariant graph neural networks with structural message-passing.
>
> [4] Zheng Wang, Xiaojun Ye, Chaokun Wang, Jian Cui, and S Yu Philip. Network embedding with completely-imbalanced labels. IEEE TKDE, 33(11), 2020.
>
> [5] Tianxiang Zhao, Xiang Zhang, and Suhang Wang. Graphsmote: Imbalanced node classification on graphs with graph neural networks. In WSDM, 2021.
>
> [6] Liang Qu, Huaisheng Zhu, Ruiqi Zheng, Yuhui Shi, and Hongzhi Yin. Imgagn: Imbalanced network embedding via generative adversarial graph networks. In KDD, 2021.
>
> [7] Seiji Maekawa, Yuya Sasaki, George Fletcher, and Makoto Onizuka. Gencat: Generating attributed graphs with controlled relationships between classes, attributes, and topology. arXiv preprint arXiv:2109.04639, 2021.
>
> [8] Thomas N. Kipf and Max Welling. Semi-supervised classification with graph convolutional networks. In ICLR, 2017.
>
> [9] Petar Velickovic, Guillem Cucurull, Arantxa Casanova, Adriana Romero, PietroLiò, and Yoshua Bengio. Graph Attention Networks. In ICLR, 2018.
>
> [10] Keyulu Xu, Weihua Hu, Jure Leskovec, and Stefanie Jegelka. How powerful are graph neural networks? ICLR, 2019.
>
> [11] Hongbin Pei, Bingzhen Wei, Kevin Chen-Chuan Chang, Yu Lei, and Bo Yang. Geom-gcn: Geometric graph convolutional networks. In ICLR, 2020.
>
> [12] Sunil Kumar Maurya, Xin Liu, and Tsuyoshi Murata. Improving graph neural networks with simple architecture design. arXiv preprint, 2021.
>
> [13] Eli Chien, Jianhao Peng, Pan Li, and Olgica Milenkovic. Adaptive universal generalized pagerank graph neural network. In ICLR, 2021.

---

> > ### Comment · Reviewer_jwKc · 2022-08-26
> > **Thank you for your clarification**
> >
> > Thank you for your answers. Based on the replies which clarifies some concerns related to the "experimental analysis, insights" nature of this work and specific contribution with respect to a previous work, I raise my score by a point.

---

> > > ### Author Response · Authors · 2022-08-27
> > > **Thank you for the response and raising your score.**
> > >
> > > We are very happy to address your questions and concerns.
> > > We would like to ask if there are any remaining concerns since your score is still tending to be negative. Could you tell us which weaknesses you pointed out still remain?

---

> ### Author Response · Authors · 2022-08-12
> **Response 1/2**
>
> Dear Reviewer jwKc,
>
> We really appreciate your comments and your support for our work. We hope our responses can address your concerns. Please find our detailed responses below.
>
> > Some existing work in this regard, such as GraphWorld [1] or the synthetic datasets used in [2], [3], among others, already use synthetic datasets for fine grained analysis or testing some particular graph properties. In such case, the use of synthetic datasets for deeper investigation is not novel to some extent.
>
> In our work, we provide deep analysis through extensive experiments using synthetic graphs generated by the state-of-the-art graph generator GenCAT, by focusing on a node classification task that is one of the hottest topics on graph neural networks.
>
> GraphWorld [1] provides limited insights for node classification with GNNs because it ignores three aspects. First, it ignores recent GNNs generalizing to heterophilic graphs, which have attracted much attention from the community. Second, the study does not care about class size distributions though they largely affect classification results. Note that a class imbalance problem has been explored in the machine learning field including graph mining [4,5,6]. Third, GraphWorld has not explored the efficiency of GNNs, which is one of the most common concerns of machine learning methods. We additionally described the difference between our work and GraphWorld in Section 7, to address this concern.
>
> Next, studies [2,3] address a task predicting graph properties such as single-source shortest-paths and Laplacian features. So, they do not provide insight regarding node classification, which is one of the hottest topics on graph neural networks.
>
> In summary, our paper provides new and interesting insights that have not been explored by existing work, by focusing on node classification from the viewpoints of detailed class structures including class size distributions and heterophily-homophily. Though it is a common approach to use synthetic datasets for deep investigation, our insights and codebase are new and novel.
>
> > For example, the case of higher or lower homophily is explored in [1]. In this regard, the contribution of this paper seems limited.
>
> As we described in the above response, GraphWorld does not utilize recent GNNs generalizing to heterophilic graphs, which have attracted much attention from the community. Also, it ignores class size distributions that can affect classification results. In contrast, our empirical studies utilize the very recent GNNs generalizing to heterophilic graphs and explore various graphs with controlled class structures. Through the experiments, we provide new insights for node classification with GNNs. Hence, the contributions of our paper are novel.
>
> > this paper seems to be an experimental demonstration, instead of a framework presented, as far as this track is concerned.
>
> The webpage of NeurIPS 2022 Datasets and Benchmarks Track (https://neurips.cc/Conferences/2022/CallForDatasetsBenchmarks) says “systematic analyses of existing systems on novel datasets that yield important new insight are also in scope.”. Actually, we present new insights from our empirical studies using various synthetic graphs suitable for the evaluation and also publicly provide an evaluation framework for future research. Hence, our work is well suited to the Datasets and Benchmarks Track.
>
> > If graph generator is somehow a contribution of the paper, it is already presented in a prior work.
>
> The contributions of our paper do not overlap those of the graph generator’s paper [7] proposing GenCAT. First, the contributions of GenCAT are two-fold; 1) the work identifies graph and class features that capture the characteristics of attributed graphs with node class labels and 2) GenCAT effectively and efficiently generates graphs with controlled characteristics by using the graph and class features.
>
> In contrast, our paper focuses on empirical studies using 16 GNN methods and various synthetic graphs suitable for their evaluation, not just a graph generation problem. Also, as we clearly described in Sections 1 and 8, our contributions are interesting insights and open questions for future research through our empirical studies. Hence, our paper makes a new contribution, rather than simply adopting prior research.
>
> > An answer to your question; “Are there particular set of references which support this statement in line 206-207?”
>
> Yes, many existing works developing new GNNs use only accuracy to evaluate their performance and we show a set of such references below [8-13].
>
>
>
> Thank you again for raising your concerns. They are really helpful to make our work better. We hope our responses sound correct for you.

---

### Official Review · Reviewer_SnKV · 2022-07-29

**Rating:** 6
**Confidence:** 4
**Clarity:** Yes, the paper is well written.

**Strengths:**

- The paper is well-organized and clearly written, with clear motivations of existing evaluations not considering varying characteristics among graphs.
- The selected graph characteristics are useful. Further study on those characteristics will be beneficial for understanding GNNs better.
- The experiments are rigorously conducted with clear details of experimental setups.

**Weaknesses:**

The evaluations designed and conducted for the proposed 4 graphs characteristics are meaningful overall, but there are several limitations that can be further improved.
- For $\rho^{\mathrm{conf}}$ and $M^{\mathrm{conf}}$, it is not easy to see how varying the hyperparameters will change those values or change how the graphs look like. Providing some simple examples (e.g. $\rho^{\mathrm{conf}}$ for each class with certain $\alpha$s, visualization of a demo graph with only <10 nodes and edges) will be highly appreciated.
- For graph size experiments, the gaps between different $(n,m)$ pairs are too small. The efficiency and performance changes of a certain method will be much more significant when it is run on two graphs largely different in sizes e.g. cora and ogbn-arxiv, revealing some meaningful limitations. I would suggest authors at least add a graph with 100000 nodes, a size of the same magnitude as ogbn-arxiv (~170000 nodes), and it is acceptable to just put on a note (e.g. OOM or unfinished after 48 hrs) for methods that are not scalable.
- It would be interesting to see how Figure 3 (different attribute-class correlations) looks with 1 or 2 more examples from other domains than citation networks to study how attribute values affect the performance i.e. changing $H^{\mathrm{Cora}}$ to $H^{\mathrm{(a \ non \ citation \ network \ dataset)}}$.


**Additional Feedback:**

As evaluated nonhomophilous GNNs do not utilize the topology information in the most heterophilic settings well, only achieving marginal improvements to MLP, it will be interesting to add **LINKX** and investigate how it performs on those synthetic graphs (especially the heterophilic settings), as it represents a simple fusion between the MLP learned node features and LINK regression learned graph topology.

**Correctness:**

The evaluation methods and experiment design appropriate and performed correctly mostly, with a few limitations mentioned in the above weakness session.

**Documentation:**

Yes, there is sufficient detail to support reproducibility.

**Relation To Prior Work:**

The literature review on GNNs for node classification, datasets and benchmarks, graph generators are comprehensive. The unique contributions of this work are highlighted clearly.


**Summary And Contributions:**

This paper proposes a new benchmark to understand the performance of GNNs on graphs with different characteristics, which are class sizes distribution, the relationship between classes and topology, the relationship between classes and attributes, and graph sizes. The authors leverage a synthetic graph generator that can generate graphs having controlled characteristics above as references to real world graphs. Rigorous and comprehensive benchmarking experiments using a variety of representative GNN architectures are conducted on those graphs, demonstrating some interesting insights with respect to graph characteristics.

---

> ### Author Response · Authors · 2022-08-12
> **Response**
>
> Dear Reviewer SnKV,
>
> We really appreciate your comments. We hope our point-to-point responses can address your concerns and clarify our contribution.
>
> > it is not easy to see how varying the hyperparameters will change those values or change how the graphs look like.
>
> To address your concern about how generated graphs look like when varying the hyperparameters, we additionally provide visualization of generated graphs for several cases in Figures 7-10 in the updated supplementary material. We hope that they help users intuitively understand how the graphs look like.
>
> > I would suggest authors at least add a graph with 100000 nodes, a size of the same magnitude as ogbn-arxiv (~170000 nodes), and it is acceptable to just put on a note (e.g. OOM or unfinished after 48 hrs) for methods that are not scalable.
>
> We conducted experiments using graphs with 60000 and 120000 nodes and reported the results in Figure 15 in the updated supplementary material. The summary of the results is two-fold: 1) the results of most GNNs on large graphs have the same tendency to those on small graphs, and 2) GNNs using a high-order adjacency matrix such as H2GCN cannot scale well since computing a high-order adjacency matrix is out-of-memory if graphs are large.
>
> > It would be interesting to see how Figure 3 (different attribute-class correlations) looks with 1 or 2 more examples from other domains than citation networks to study how attribute values affect the performance
>
> We agree that experiments using various domains provide more generalizability. However, we only use a single dataset, Cora which is one of the most widely used datasets, due to time limitation. Adding a new basis dataset for synthetic graphs is our future work.
>
> > it will be interesting to add LINKX and investigate how it performs on those synthetic graphs
>
> As you mentioned in your review comments, the very recent method LINKX is interesting to investigate its performance on synthetic graphs. So, we have additionally conducted experiments for LINKX and reported the results in Figures 1-6, and 11-14. Our conclusion regarding LINKX is following: LINKX works in the both homophily and heterophily settings. However, it cannot achieve the state-of-the-art f1-macro scores. This is because we just follow the hyperparameter search space specified in [1] (for details, see the hyperparameter search space described in the updated supplementary material). Broader search space may increase the classification performance.
>
> Thank you again for giving us helpful feedback and suggestions. We hope that the additional experiments make our observations and insights more valid and our responses can fully address your concerns.
>
> **References**
>
> [1] Derek Lim, Felix Hohne, Xiuyu Li, Sijia Linda Huang, Vaishnavi Gupta, Omkar Bhalerao, and Ser Nam Lim. Large scale learning on non-homophilous graphs: New benchmarks and strong simple methods. In NeurIPS, 2021.

---

> > ### Comment · Reviewer_SnKV · 2022-08-27
> > **Thank you for the detailed response**
> >
> > Thank you for your response. I incline to accpet this paper, but I think my current score is reasonable. Thus I will keep my score as it is.

---

> > > ### Author Response · Authors · 2022-08-29
> > > **Thank you for your reply.**
> > >
> > > Thank you for your careful comments and feedback again. They are really helpful to improve our paper. For example, adding visualization of generated graphs used in the experiments and conducting additional experiments for larger graphs and a very recent method LINKX.
> > > We would like to ask if there are any remaining concerns.

---

### Author Response · Authors · 2022-08-23
**Are there any remaining concerns or open questions?**

Dear reviewers and area chairs,

It has taken 10 days since we posted our responses to reviewers' comments and one week is left for the discussion period. We would like to ask if there are any remaining concerns and open questions. If reviewers' concerns have been addressed, we thank the reviewers again and look forward to the final decision.

---

### Meta-Review · Area_Chair_X6GC · 2022-09-04

**Recommendation:** Accept
**Confidence:** 4

**Metareview:**

This paper proposes a benchmark for evaluating node classification of GNNs, considering several characteristics of the graph, including the class size distribution, edge connection proportion, attribute value, and graph size. The concerns lie in that this paper mainly builds the synthetic graph datasets without very significant differences or new insights from existing works. The motivations for "beyond real-world benchmark" stated in the title are also not very clear. I expect the author(s) could well address these concerns to improve this paper.

All in all, I recommend a borderline accept if there is room for the proceedings.

---

### Decision · Program_Chairs · 2022-09-16

Accept